# Telomerase biogenesis requires a novel Mex67 function and a cytoplasmic association with the Sm$_7$ complex

Yulia Vasianovich[†], Emmanuel Bajon[‡], Raymund J Wellinger*

Department of Microbiology and Infectious Diseases, Faculty of Medicine and Health Sciences, Université de Sherbrooke, Sherbrooke, Canada

**Abstract** The templating RNA is the core of the telomerase reverse transcriptase. In *Saccharomyces cerevisiae*, the complex life cycle and maturation of telomerase includes a cytoplasmic stage. However, timing and reason for this cytoplasmic passage are poorly understood. Here, we use inducible RNA tagging experiments to show that immediately after transcription, newly synthesized telomerase RNAs undergo one round of nucleo-cytoplasmic shuttling. Their export depends entirely on Crm1/Xpo1, whereas re-import is mediated by Kap122 plus redundant, kinetically less efficient import pathways. Strikingly, Mex67 is essential to stabilize newly transcribed RNA before Xpo1-mediated nuclear export. The results further show that the Sm$_7$ complex associates with and stabilizes the telomerase RNA in the cytoplasm and promotes its nuclear re-import. Remarkably, after this cytoplasmic passage, the nuclear stability of telomerase RNA no longer depends on Mex67. These results underscore the utility of inducible RNA tagging and challenge current models of telomerase maturation.

*For correspondence:
Raymund.Wellinger@
Usherbrooke.ca

Present address: [†]Department of Medicine, Faculty of Medicine and Health Sciences, Research Institute of the McGill University Health Center (RI-MUHC), McGill University, Montreal, Canada; [‡]Department of Biochemistry, Faculty of Medicine, Université de Montréal, Montreal, Canada

## Introduction

Telomeres are essential nucleoprotein structures that protect linear eukaryotic chromosome ends from detrimental repair processes as well as natural replication-associated shortening (*Hayflick, 1965*; *Harley et al., 1990*; *Wellinger and Zakian, 2012*; *de Lange, 2018*). In stem, germ, and most cancer cells, the enzyme telomerase maintains telomeric DNA at a functional length, which is key for the unlimited proliferation capacity of these cells (*Shay, 2016*; *Roake and Artandi, 2020*). Telomerase synthesizes telomeric repeats *de novo* using a reverse transcriptase activity and its internal RNA as a template (*Greider and Blackburn, 1985*; *Greider and Blackburn, 1987*). Unicellular eukaryotes such as yeasts express telomerase constitutively. Notably, the overall telomerase composition and aspects of telomerase dynamics are strikingly conserved between yeast and higher eukaryotes (*Vasianovich and Wellinger, 2017*). This makes yeast a great model organism to study telomerase behavior and for predicting this process in higher eukaryotes.

Telomerase is a complex ribonucleoprotein machine. The non-coding RNA, called TLC1 in *Saccharomyces cerevisiae*, is at the heart of the telomerase holoenzyme. It harbors a template for telomeric DNA synthesis and acts as a scaffold for telomerase-associated proteins (*Wellinger and Zakian, 2012*). The repertoire of the budding yeast telomerase proteins includes the Est2 catalytic subunit and several additional proteins such as Est1, Est3, Pop1, Pop6, Pop7, the Sm$_7$, and Yku70/Yku80 complexes, which all are required for telomerase function *in vivo*. Like most yeast snRNAs, the major form of TLC1 is a poly-A$^-$ RNA Pol II transcript, which has a TMG cap at the 5'-end and associates with the ring-shaped heptameric Sm$_7$ complex at its 3'-end (*Chapon et al., 1997*; *Seto et al., 1999*). This Sm$_7$ complex is crucial for RNA maturation, including 5'-TMG capping and 3'-end processing, and is absolutely essential for the stability of both telomerase and snRNAs (*Rymond, 1993*; *Roy et al., 1995*; *Seto et al., 1999*; *Coy et al., 2013*; *Shukla and Parker, 2014*).

TLC1 is an extremely low abundance RNA. According to different estimates, only 10–30 mature TLC1 molecules are present in a haploid yeast cell, thus severely limiting the quantity of functional telomerase complexes (*Mozdy and Cech, 2006*; *Gallardo et al., 2008*; *Bajon et al., 2015*). As a result, the 64 chromosome ends present in a haploid cell after DNA replication must rely on a very limited number of telomerase RNPs for their maintenance. In fact, the available nuclear telomerase arsenal may be even lower, as at steady state, 30–40% of TLC1 molecules reside in the cytoplasm (*Gallardo et al., 2008*). However, the reason for the cytoplasmic localization of telomerase RNA is still unclear.

One hypothesis stipulates that the TLC1 RNA shuttles to the cytoplasm as a part of its biogenesis process (*Ferrezuelo et al., 2002*; *Gallardo et al., 2008*). According to this model, TLC1 gets exported to the cytoplasm to associate with some of its protein subunits, followed by re-import of the RNP to the nucleus. However, which telomerase subunits bind to TLC1 in the cytoplasm is unclear. Furthermore, the current view of telomerase biogenesis suggests that the 3'-stabilizing $Sm_7$ ring binds telomerase RNA very early in its life cycle, prior to RNA export to the cytoplasm (*Gallardo et al., 2008*). Consistently, the SmB, D1, and D3 subunits of the $Sm_7$ complex contain NLS sequences, suggesting that the ring can get imported into the nucleus on its own (*Bordonné, 2000*). In addition, TMG capping, which requires $Sm_7$, appears functional when TLC1 export is blocked (*Gallardo et al., 2008*). Finally, since the $Sm_7$ complex is crucial for the stability of telomerase RNA (*Seto et al., 1999*), it would be expected to associate with TLC1 at the earliest possible stages. However, in both yeast and mammals, snRNAs that belong to the $Sm_7$ class are exported to the cytoplasm to assemble with the $Sm_7$ complex (*Fischer et al., 2011*; *Becker et al., 2019*). Therefore, if TLC1 follows a path similar to snRNAs, newly synthesized telomerase RNA may temporarily shuttle to the cytoplasm to assemble with the $Sm_7$ proteins, which would contradict the above model. In addition, the cytoplasm may also be the final destination for TLC1 molecules, where old dysfunctional telomerase RNAs would be degraded. Therefore, the cytoplasmic TLC1 fraction may be a complex mix of both old degrading molecules as well as newly synthesized RNA transcripts that undergo assembly with protein subunits.

There is evidence that nucleo-cytoplasmic export of telomerase RNA is mediated by the Xpo1/Crm1 exportin and re-import is dependent on Mtr10 as well as Kap122 (*Ferrezuelo et al., 2002*; *Gallardo et al., 2008*). However, molecular details of TLC1 transport by these factors are missing, for example it is unclear at what stage and in which state TLC1 may shuttle to the cytoplasm or whether it gets exported permanently for degradation.

The Xpo1/Crm1 exportin belongs to the family of karyopherins which transport cargo RNA and proteins through nuclear pores using a RanGTP – RanGDP gradient (*Köhler and Hurt, 2007*). Many RNA species, such as snRNAs, rRNAs, and tRNAs depend on Xpo1 for their export. In contrast, transport of mRNA relies on the Mex67 export receptor (*Segref et al., 1997*; *Santos-Rosa et al., 1998*), which is structurally unrelated to karyopherins and does not depend on a RanGTP gradient (*Köhler and Hurt, 2007*). Inactivation of Mex67 leads to nuclear accumulation of mRNA (*Segref et al., 1997*; *Santos-Rosa et al., 1998*). In addition, in the absence of Mex67, poly-A$^+$ RNA appears to be extremely unstable, which was attributed to untimely RNA decay due to the export block (*Tudek et al., 2018*). Notably, Mex67 is also somehow implicated in the cytoplasmic export of Xpo1-transported cargoes, such as snRNA, rRNA, and tRNA, raising a question of why these RNA species require two export receptors (*Yao et al., 2007*; *Faza et al., 2012*; *Chatterjee et al., 2017*; *Becker et al., 2019*). In fact, the precise function of Mex67 and the molecular details of its RNA export role are unknown.

In this study, we aimed to dissect the dynamic, life-cycle-dependent localization of telomerase RNA molecules via single molecule fluorescence *in situ* hybridization (smFISH) microscopy. For this purpose, we required a tool that would discriminate and selectively visualize new *vs* pre-existing matured telomerase RNA transcripts within the total TLC1 population. Fluorescent imaging of telomerase RNA molecules tagged with the MS2 repeats has been successfully used in the past (*Gallardo et al., 2011*; *Cusanelli et al., 2013*; *Bajon et al., 2015*). However, constitutive RNA tagging has its limitations: it captures the general RNA population at its steady state and therefore cannot reveal the dynamics of specific RNA subpopulations, such as new or old TLC1 molecules. In order to identify these fractions in the total TLC1 population and to reveal their specific behavior, we converted the constitutive TLC1-MS2 tagging system into an inducible one using a recombination-induced tag exchange approach (*Verzijlbergen et al., 2010*). In this system, controlled

induction of TLC1 tagging with the MS2 repeats results in gradual appearance of *new* TLC1-MS2 molecules, which can be easily distinguished from the bulk untagged TLC1 population using an MS2-specific FISH probe. At the same time, untagged TLC1 molecules that lack the MS2-specific signal will represent the pre-existing TLC1 fraction which for simplicity will be referred to as *old* TLC1 molecules.

Using this inducible TLC1 tagging tool, we show that very rapidly after transcription, newly produced telomerase RNA molecules are temporarily exported to the cytoplasm, consistent with the shuttling model. Old TLC1 transcripts also tend to accumulate in the cytoplasm likely for degradation purposes. Therefore, the results show that the cytoplasmic TLC1 RNA is *per se* a heterogeneous population comprised of both new and old telomerase RNA transcripts. The data also show that Xpo1 and Kap122, previously shown to influence subcellular TLC1 localization, affect the shuttling of newly synthesized TLC1 molecules. Notably, cytoplasmic export of TLC1 is entirely dependent on Xpo1, whereas telomerase RNA re-import to the nucleus seems to involve not only Kap122 but also other less efficient import mechanisms. Strikingly, Mex67 plays an entirely unexpected role in the biogenesis of telomerase RNA. While cells without the Xpo1 exportin sequester all newly synthesized TLC1 in the nucleus, disruption of Mex67 leads to a drastically different phenotype, resulting in a complete loss of new TLC1 transcripts. This phenotype is recapitulated in a *mex67-5 xpo1-1* double mutant and, remarkably, is suppressed by inactivation of the nuclear exosome subunit *RRP6*. Thus, Mex67 is crucial for stabilization of telomerase RNA in the nucleus immediately after transcription and prior to export – a function that may be required for other RNA species as well. Finally, we show that similar to snRNAs, telomerase RNA assembles with the $Sm_7$ complex in the cytoplasm. This association is essential for the stability of TLC1 RNA in the cytoplasm and its re-import to the nucleus. Cytoplasmic association of telomerase RNA with the $Sm_7$ complex implies that the downstream $Sm_7$-dependent events such as 5'- and 3'-end RNA processing occur after TLC1 returns to the nucleus. Therefore, in this study we challenge the current view of telomerase RNA biogenesis by adding Mex67 as a novel player essential for nuclear TLC1 stability right after transcription, and by localizing the $Sm_7$-binding step to the cytoplasm after the export of TLC1. These results underscore the general utility and validity of an inducible tagging system to probe dynamic localization details during the life cycle of any RNA.

## Results

### Inducible *TLC1-[MS2-IN]* tagging system

The MS2 tag used in earlier studies for constitutive TLC1 RNA tagging comprises 10 MS2 stem loops in the proximity to the TLC1 3'-end, just upstream of the $Sm_7$ binding site (*Gallardo et al., 2011*; *Bajon et al., 2015*). To convert this construct into an inducible tagging system (named *TLC1-[MS2-IN]*), a recombination cassette was placed upstream of the *10xMS2* array (*Figure 1A*). The recombination cassette contained the *TLC1* 3'-end, the *natMX4* selection marker and was flanked by the *loxP* sites. The *TLC1-[MS2-IN]* construct was integrated in the *TLC1* genomic locus to allow for endogenous RNA expression levels. Under normal conditions, wild type TLC1 RNA transcripts are expressed from the *TLC1-[MS2-IN]* locus (*Figure 1B*). However, induction of recombination by Cre-recombinase will lead to the excision of the fragment flanked by the *loxP* sites and join the *MS* array sequence downstream to the *TLC1* 3'-end (*Figure 1A,B*). As a result, tagged TLC1-MS2 molecules will start to be transcribed from the recombined *TLC1-[MS2-IN]* locus. In addition, a non-replicating episome containing the selection marker and the wild type *TLC1* 3'-end will be produced upon recombination (*Figure 1A*). To induce recombination, we used a constitutively expressed Cre-recombinase fused to the estradiol-binding domain (Cre-EBD). Cre-EBD enters the nucleus only in the presence of β-estradiol, which allows for rapid and controlled induction of *TLC1-[MS2-IN]* recombination (*Logie and Stewart, 1995*).

After addition of β-estradiol to the exponentially growing cultures, recombination of the *TLC1-[MS2-IN]* genomic locus was followed for 16 hr by Southern blotting (*Figure 1C,D*). In addition, appearance of tagged TLC1-MS2 RNA transcripts was monitored by northern blotting and RT-ddPCR (*Figure 1E–G*). During the experiment, a cloned strain with the recombined *TLC1-[MS2-IN]* locus was used as a reference to monitor the efficiency of recombination and TLC1-MS2 tagging. Southern blot analysis showed that *TLC1-[MS2-IN]* recombination occurred very rapidly, reaching

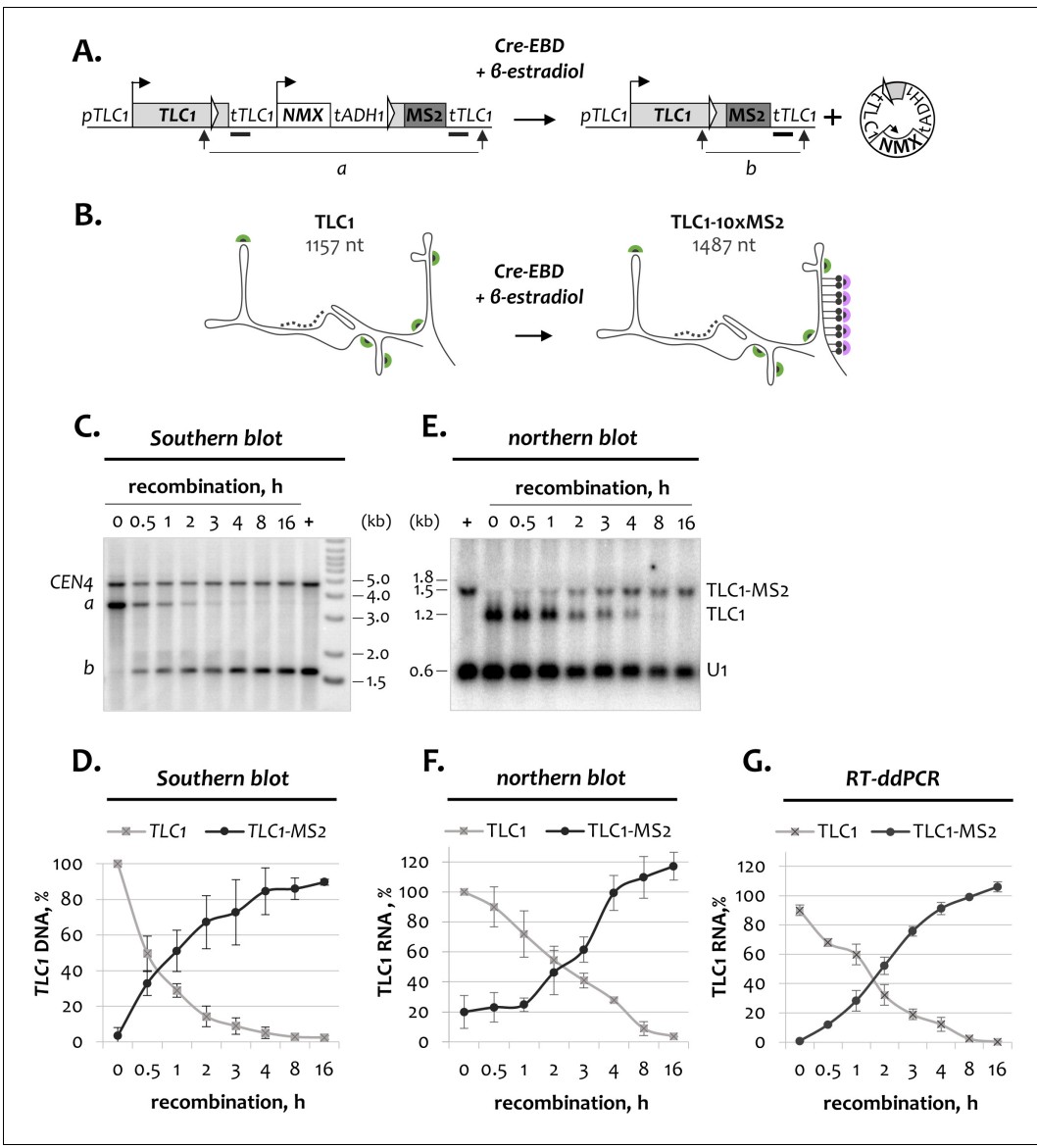

**Figure 1.** Kinetics of inducible TLC1-MS2 tagging. (**A**) Schematic of the *TLC1-[MS2-IN]* DNA locus before and after recombination. Thick black bars indicate the TLC1 probe used for Southern blot analysis in C. Arrows mark restriction fragments (a and b) detected by the TLC1 probe. *NMX – natMX4* cassette. (**B**) Schematic of tagged and untagged TLC1 RNA before and after recombination, respectively. Dotted line indicates the TLC1 probe used for northern blot analysis in E. Green and purple half-circles indicate TLC1- and MS2-specific smFISH probes, respectively. (**C**) Southern blot analysis of the *TLC1-[MS2-IN]* locus following induction of recombination. The position of the TLC1 probe hybridizing to restriction fragments a and b is depicted in A. *CEN4* – loading control. '+' - positive recombined control. DNA marker sizes are indicated on the right (kb). (**D**) Quantification of the Southern blot shown in C. Fractions relative to 0 hr (for unrecombined *TLC1* locus) and to the positive recombined control (+) (for recombined *TLC1-MS2* locus) are shown. Error bars indicate SD, n = 3. (**E**) Northern blot analysis of untagged TLC1 and tagged TLC1-MS2 RNA transcripts following induction of *TLC1-[MS2-IN]* recombination. Position of the TLC1 probe used to detect TLC1 species is shown in B. U1 – loading control. '+' - positive recombined control. RNA sizes are indicated on the left (kb), 1.8 kb corresponds to 18S rRNA. (**F**). Quantification of the northern blot shown in E. Fractions relative to 0 hr (for untagged TLC1) and to the positive recombined control (+) (for tagged TLC1-MS2) are shown. Error bars indicate SD, n = 3. (**G**) RT-ddPCR quantification of untagged TLC1 and tagged TLC1-MS2 RNA transcripts following induction of recombination. Fractions relative to the positive recombined control are shown. Error bars indicate SD, n = 2.

The online version of this article includes the following source data for figure 1:

**Source data 1.** Numerical data used to generate *Figure 1*.

almost 50% after 30 min of β-estradiol addition to the culture (*Figure 1C,D*). Conversion of the *TLC1-[MS2-IN]* locus into *TLC1-MS2* was nearly complete after 4 hr, reaching approximately 90% relative to the reference strain. At the RNA level, we also observed gradual disappearance of the shorter untagged TLC1 molecules and appearance of the longer TLC1-MS2 RNA species (*Figure 1E, F*). After 2.5 hr of recombination induction, nearly half of TLC1 molecules had acquired a tag, whereas by the end of the time course, the TLC1 population consisted almost exclusively of tagged TLC1-MS2 molecules. Independent quantification of tagged and untagged TLC1 species throughout the experiment by RT-ddPCR yielded results consistent with the northern blot analysis (*Figure 1G*). Therefore, the inducible recombination system constructed in this study is functional and allows TLC1-MS2 tagging in a fast and controlled manner.

## Tracking subcellular distribution of telomerase RNA fractions

The turnover of the TLC1 RNA population was also monitored by the two-color smFISH analysis. For this approach, we used a combination of TLC1-specific probes that detect all TLC1 RNA species, and MS2-specific probes that hybridize only to newly tagged molecules (*Figure 1B*). The specificity of the probes was confirmed using a *tlc1Δ* control strain, that lacked either TLC1 or MS2 FISH signals (*Figure 2—figure supplement 1A*).

Before recombination induction, *TLC1-[MS2-IN]* cells contained only TLC1-specific foci, indicating that only untagged TLC1 RNA was expressed at the 0 hr time point (*Figure 2A*). At this stage (called steady state), the average number of TLC1 RNA was ~10 molecules/cell, which is consistent with previous reports (*Figure 2B*, *Figure 2—figure supplement 1B*; *Gallardo et al., 2008*; *Bajon et al., 2015*). After induction of recombination, old foci hybridizing only with TLC1 probes slowly disappeared, whereas new foci hybridizing with both TLC1 and MS2 probes gradually accumulated (*Figure 2A,B*, *Figure 2—figure supplement 1B,C*). The first MS2-specific foci appeared as early as 30 min after β-estradiol addition (*Figure 2A,B*, *Figure 2—figure supplement 1C*). By the end of the 16 hr time course, the tagged TLC1-MS2 fraction constituted more than 80% of total observed TLC1 molecules (*Figure 2B,C*). However, a small fraction of TLC1 foci remained untagged even after 16 hr of recombination. These might be attributed to late recombination events that occur in a sub-population of cells (*Figure 1D*).

In order to verify that the *TLC1-[MS2-IN]* system does not disrupt the telomerase RNA function or RNP biogenesis, we performed telomere length analysis before and after recombination (*Figure 2—figure supplement 1D*). Consistent with previous reports, clones with the recombined *TLC1-MS2* locus had stable and functional telomeres that were only slightly shorter than wild type telomeres (*Gallardo et al., 2011*; *Bajon et al., 2015*). *TLC1-MS2* clones isolated after recombination induction also had a slightly reduced number of TLC1 molecules compared to wild type cells (*Figure 2—figure supplement 1E*, *top*). At present we do not know the reason for this minor reduction of RNP abundance, but virtually all functions of the enzyme, and in particular its enzymatic activity, remain unaffected by the tag (*Gallardo et al., 2011*; *Bajon et al., 2015*). Importantly, the nucleo-cytoplasmic distribution of TLC1 foci remained unchanged after inducible TLC1 tagging, validating that our system is suitable to track subcellular movements of old and new TLC1 molecules separately (*Figure 2—figure supplement 1E*, *bottom*).

At steady state (0 hr time point), the majority of TLC1 molecules resided in the nucleus (60–70%), whereas the remaining 30–40% localized in the cytoplasm (*Figure 2D*). However, this distribution started to change after induction of recombination. Over time, old untagged TLC1 molecules gradually accumulated in the cytoplasm, whereas their nuclear fraction declined. After 16 hr of recombination, the majority of old TLC1 molecules resided in the cytoplasm, as opposed to predominantly nuclear TLC1 localization at steady state (*Figure 2D*). Hence, old telomerase RNA accumulates in the cytoplasm at the end of its life cycle. Nevertheless, even by the end of the time course, a significant fraction of the remaining old TLC1 molecules still localized in the nucleus (*Figure 2D*).

Quantification of the subcellular distribution of new TLC1-MS2 molecules revealed a peculiar behavior. In contrast to steady state, early in the time course, i.e. immediately after RNA transcription, new TLC1-MS2 molecules localized predominantly in the cytoplasm (*Figure 2E*). This behavior persisted for approximately 2 hr, followed by gradual return of new molecules to the nucleus and reestablishment of steady state at the end of the time course. The apparent slow dynamics of TLC1 relocalization to the nucleus could also be due to delayed recombination events that occur in a fraction

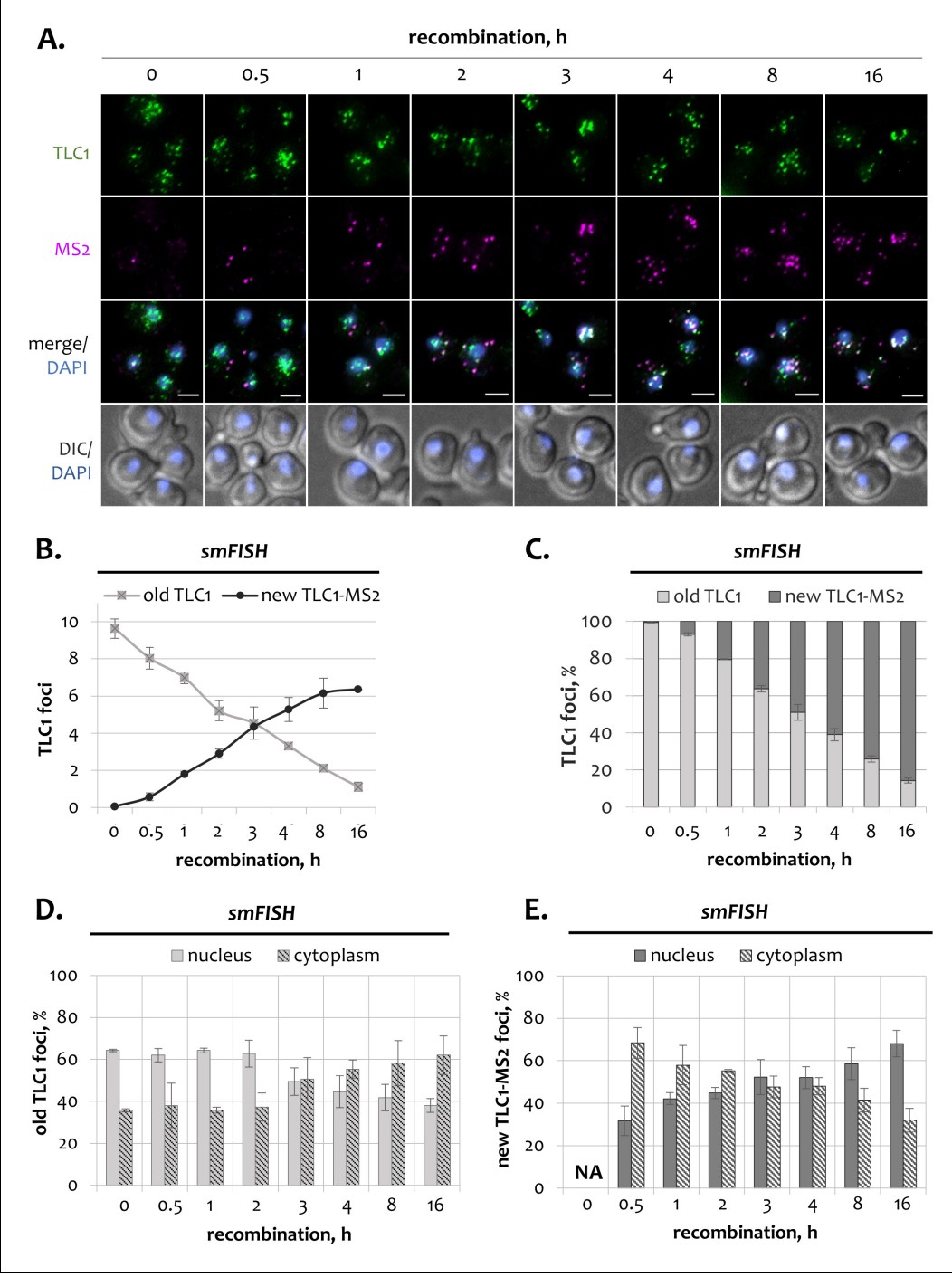

**Figure 2.** Subcellular distribution of telomerase RNA fractions in wild type cells. (**A**) Two-color smFISH analysis of untagged old TLC1 (*green*) and tagged new TLC1-MS2 molecules (*purple*) in wild type cells following induction of recombination. TLC1-Cy3 and MS2-Cy5 probes were used. Scale bar, 2 μm. (**B**) Average number of old and new TLC1 foci observed by smFISH following recombination induction. 50–100 cells from three independent experiments were scored. Error bars indicate SD. (**C**) Relative fractions of old and new TLC1 foci observed by smFISH following recombination induction. 50–100 cells from three independent experiments were scored. Error bars indicate SD. (**D, E**). Relative nuclear and cytoplasmic fractions of old (**D**) and new (**E**) TLC1 foci observed by smFISH following recombination induction. NA – not applicable. 50–100 cells from three independent experiments were scored. Error bars indicate SD.

The online version of this article includes the following source data and figure supplement(s) for figure 2:

**Source data 1.** Numerical data used to generate *Figure 2*.

*Figure 2 continued on next page*

*Figure 2 continued*

**Figure supplement 1.** Additional characteristics of the *TLC1-[MS2-IN]* system.

**Figure supplement 1—source data 1.** Numerical data used to generate *Figure 2—figure supplement 1*.

of asynchronous culture (*Figure 1D*). We conclude that at early stages after transcription, TLC1 molecules are exported to the cytoplasm but eventually shuttle back to the nucleus.

In summary, the inducible TLC1 tagging system in combination with the two-color smFISH approach allows to study and quantify in detail the behavior of new and old TLC1 molecules as separate fractions. Old molecules tend to accumulate in the cytoplasm where they are likely to be degraded. On the other hand, the majority of newly synthesized telomerase RNA temporarily localizes in the cytoplasm followed by their gradual return to the nucleus and re-establishment of steady state.

## Xpo1 and Kap122 are involved in nucleo-cytoplasmic shuttling of newly synthesized TLC1 transcripts

The Xpo1 exportin and Kap122 importin were previously implicated in the subcellular distribution of TLC1 (*Gallardo et al., 2008*). In order to establish whether Xpo1 and Kap122 are required for the shuttling of newly synthesized TLC1 molecules, we analyzed the dynamic localization of tagged TLC1 in the absence of these proteins.

First, we tested whether export of new TLC1 molecules to the cytoplasm is dependent on the Xpo1 exportin (*Figure 3A,B*). For this purpose, cells harboring a temperature-sensitive *xpo1-1* allele were shifted to the restrictive temperature (37°C) 1 hr before recombination induction. After addition of β-estradiol, *xpo1-1* mutants were monitored for additional 8 hr at 37°C, at which point cells had completely stopped dividing. In *xpo1-1* cells kept at the permissive temperature, new TLC1 molecules followed similar localization dynamics as in wild type cells, temporarily localizing in the cytoplasm at early stages and eventually flowing back to the nucleus (*Figure 3—figure supplement 1C*). However, in *xpo1-1* cells incubated at the restrictive temperature, a drastic accumulation of newly transcribed TLC1 molecules in the nucleus was observed (*Figure 3A,B*). A slight increase in the cytoplasmic TLC1-MS2 fraction early in the time course (*Figure 3B*, 1 h) could be explained by incomplete inactivation of Xpo1. The accumulation of new TLC1 molecules in the nucleus observed in *xpo1-1* cells at 37°C was not due to the increase in temperature, as new TLC1 molecules in wild type cells grown at 37°C reached a normal steady state distribution, though with a slightly faster dynamics (*Figure 3—figure supplement 1D*). These data directly demonstrate that the early temporary export of newly synthesized TLC1 molecules to the cytoplasm depends on the Xpo1 exportin. Moreover, Xpo1-mediated transport is the only mechanism responsible for the export of new telomerase RNA, as upon Xpo1 inactivation virtually all TLC1 transcripts are retained in the nucleus for up to 8 hr (*Figure 3B*). Notably, in *xpo1-1* mutants incubated at 37°C but not at 25°C, a higher amount of new TLC1-MS2 RNA was detected as compared to wild type cells (*Figure 3D*; *Figure 3—figure supplement 1F*). Although *xpo1-1* mutants grown at 37°C initially have a faster recombination kinetics, this cannot explain the increased amount of TLC1 RNA at the end of the experiment (*Figure 3—figure supplement 1A,B*). Consistent with the northern blot quantification, we also observed a larger number of new TLC1 foci in *xpo1-1* mutant cells at the restrictive temperature (*Figure 3E*). Again, this effect is due to inactivation of Xpo1 rather than the elevated temperature, as wild type cells grown at 30 and 37°C have comparable numbers of new TLC1 foci (*Figure 3—figure supplement 1G*). These data show that in the absence of functional Xpo1 when the cytoplasmic export of TLC1 is completely blocked, newly made telomerase RNA persists in the nucleus and accumulates to higher levels than in export-competent cells.

To test if Kap122 is involved in the nuclear re-import of new TLC1 molecules, we followed telomerase RNA via inducible tagging in *kap122Δ* cells (*Figure 3A,C*). For the first 60 min after recombination induction in *kap122Δ* mutants, newly synthesized TLC1 RNA behaved similarly to wild type cells, localizing predominantly in the cytoplasm (*Figures 2E* and *3C*). In wild type cells, the nuclear TLC1 fraction started to significantly increase thereafter, being almost equal to the cytoplasmic level after 2 hr and predominant after 3 hr (*Figure 2E*). However, in *kap122Δ* cells, the nuclear fraction began to recuperate only 3 hr after recombination induction and never reached the wild type level, always

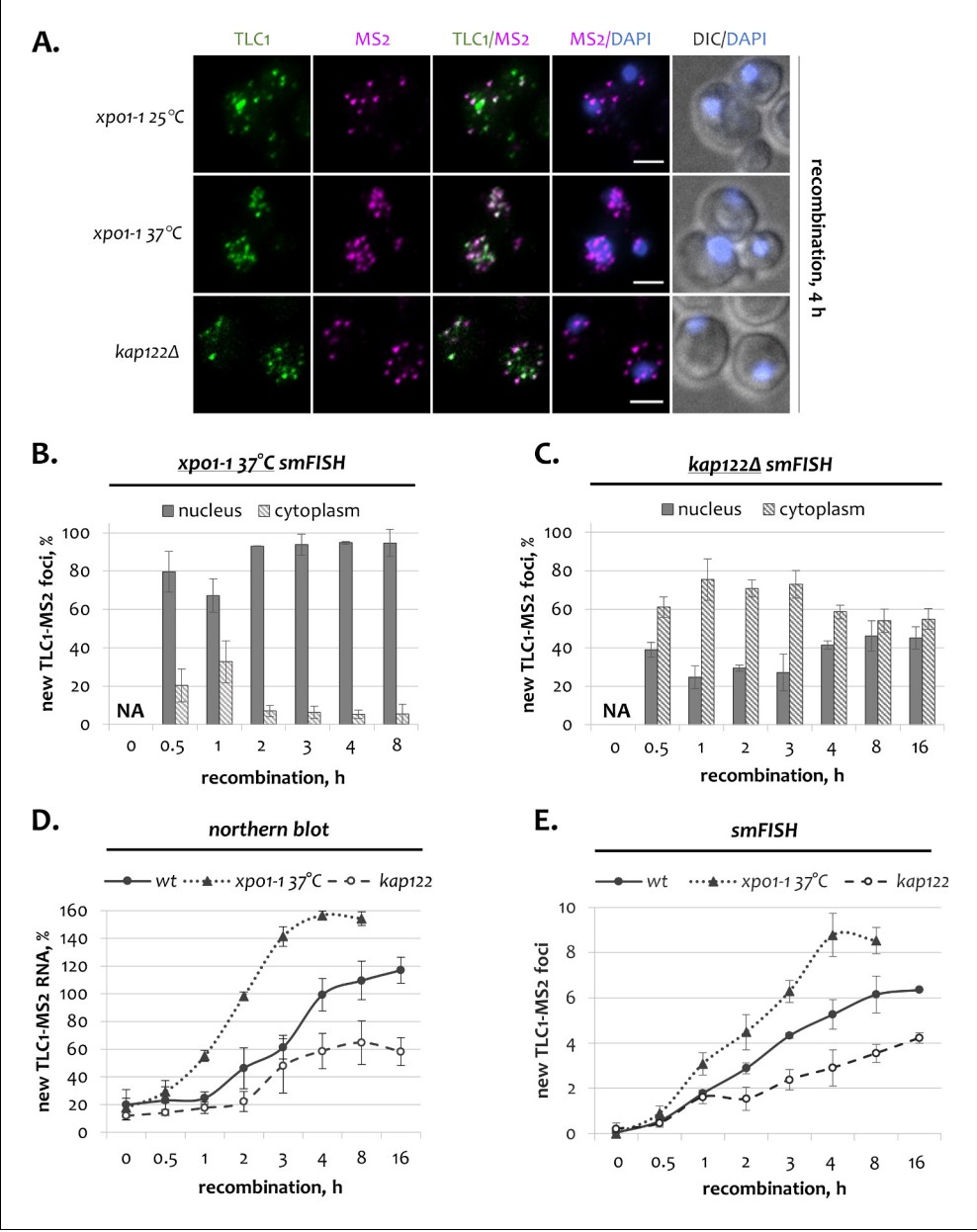

**Figure 3.** Dynamics of newly transcribed telomerase RNA in the absence of Xpo1 and Kap122. (**A**) Two-color smFISH analysis of untagged TLC1 and tagged TLC1-MS2 molecules in *xpo1-1* and *kap122* mutants after 4 hr of recombination. TLC1-Cy3 and MS2-Cy5 probes were used. Scale bar, 2 μm. (**B, C**). Relative nuclear and cytoplasmic fractions of new TLC1 foci observed by smFISH following recombination induction in *xpo1-1* (**B**) and *kap122* (**C**) mutants. NA – not applicable. 50–100 cells from three independent experiments were scored. Error bars indicate SD. (**D**). Northern blot quantification of untagged TLC1 and tagged TLC1-MS2 RNA transcripts following recombination induction in wild type, *xpo1-1* and *kap122* cells. Fractions relative to the positive recombined control are shown. Error bars indicate SD, n = 3. (**E**) Average number of new TLC1 foci observed by FISH following recombination induction in wild type, *xpo1-1* and *kap122* cells. 50–100 cells from three independent experiments were scored. Error bars indicate SD.

The online version of this article includes the following source data and figure supplement(s) for figure 3:

**Source data 1.** Numerical data used to generate *Figure 3*.
**Figure supplement 1.** TLC1 dynamics in wild type, *xpo1-1* and *kap122* cells.
**Figure supplement 1—source data 1.** Numerical data used to generate *Figure 3—figure supplement 1*.

remaining below 50% of total new molecules (*Figure 3C*). This effect is not due to slower progress of recombination, as the recombination kinetics in *kap122Δ* cells was comparable to wild type (*Figure 3—figure supplement 1A,B*). In fact, by the end of the time course, new TLC1 molecules had reached the steady state nucleo-cytoplasmic distribution that was characteristic of *kap122Δ* mutants before recombination induction (*Figures 3C*, 16 h; *Figure 3—figure supplement 1E*). Therefore, these results show that Kap122 supports the rapid re-import of new molecules to the nucleus after the cytoplasmic stage. However, the late re-import of new TLC1 into the nucleus in *kap122Δ* mutants suggests the existence of a secondary RNA import pathway. Indeed, telomere length in *kap122Δ* mutants with steady state distribution of TLC1 molecules is only slightly shorter than in wild type cells (*Figure 2—figure supplement 1D*).

Also, if the cytoplasmic telomerase RNA does not return to the nucleus in a timely fashion in the absence of Kap122, it may become subject to degradation. As a result, even if the nuclear fraction is held low, the nucleo-cytoplasmic TLC1 ratio in *kap122Δ* cells will increase over time. Consistent with this scenario, the level of new TLC1-MS2 RNA in *kap122Δ* mutants is about 2-fold lower than in wild type cells, as analyzed by northern blotting (*Figure 3D*). Consistently, a similar reduction in the number of new TLC1-MS2 foci in the absence of Kap122 was also observed by smFISH (*Figure 3E*). These data indicate that new cytoplasmic TLC1 molecules, which due to the absence of Kap122 are not rapidly transported back to the nucleus, become prone to degradation. As a result, the total quantity of telomerase RNA decreases. Therefore, while a redundant secondary TLC1 import pathway may exist, it is less efficient than the primary Kap122 pathway as it cannot prevent TLC1 losses in the cytoplasm.

## The nuclear stability of new telomerase RNA transcripts requires Mex67

Transport of mRNAs, pre-ribosomal subunits and tRNA to the cytoplasm depends on the Mex67 export receptor (*Segref et al., 1997*; *Santos-Rosa et al., 1998*; *Yao et al., 2007*; *Faza et al., 2012*; *Chatterjee et al., 2017*). In addition, Mex67 is required for the nuclear export of snRNAs and may be implicated in TLC1 transport (*Wu et al., 2014*; *Becker et al., 2019*). Therefore, we tested whether Mex67 affected the dynamics of newly synthesized TLC1 molecules. For this purpose, we performed inducible TLC1 tagging experiment in a temperature-sensitive *mex67-5* mutant. Similar to *xpo1-1* cells, *mex67-5* mutants were pre-incubated at the restrictive temperature (37°C) for 1 hr, treated with β-estradiol and monitored for 8 hr at 37°C. *mex67-5* cells grown at the permissive temperature (25°C) were used as a control. Northern blot quantification showed efficient accumulation of new TLC1-MS2 RNA at 25°C, which after 8 hr of recombination induction reached 80% relative to the recombined reference strain (*Figure 4A,B*). Steady appearance of new TLC1 foci at the permissive temperature was also detected by smFISH (*Figure 4C,D*). Surprisingly, at the restrictive temperature, new TLC1 RNA was almost undetectable either by northern blot or smFISH (*Figure 4A–D*). This absence of new TLC1 molecules was not due to a defect in *TLC1-[MS2-IN]* recombination, as the kinetics of recombination was the same both at 25 and 37°C (*Figure 4—figure supplement 1A*). In addition, untagged TLC1 RNA gradually disappeared at 37°C at a rate that was only slightly faster than at 25°C (*Figure 4—figure supplement 1B*). Therefore, these results suggest that in the absence of Mex67, new TLC1 molecules are either not produced, i.e. the gene is not transcribed, or they are extremely unstable.

To distinguish between these two scenarios, we combined the temperature shift and a transcription block into a single experiment (*Figure 4F*; *Figure 4—figure supplement 1D*). First, *mex67-5* cells were treated with β-estradiol for 4 hr at the permissive temperature to allow proper formation of new TLC1 molecules (*Figure 4F*). During this stage, accumulation of new TLC1 molecules in *mex67-5* cells was comparable to wild type. After shifting the temperature to 37°C, production of new TLC1 molecules continued in wild type cells, but stopped abruptly in *mex67-5* mutants, as expected from the above (*Figure 4F*; *Figure 4—figure supplement 1D*). If the dysfunctional Mex67 at the restrictive temperature abrogated TLC1 transcription, then an additional transcription block would not be expected to aggravate the *mex67-5* phenotype. However, addition of the RNA polymerase inhibitor thiolutin before shifting the temperature to 37°C resulted in an even further loss of new TLC1 molecules (*Figure 4F*). These data indicate that in *mex67-5* cells incubated at the restrictive temperature, TLC1 transcription is functional. In addition, the level of RNA Pol II detected at the *TLC1* locus was comparable in *mex67-5* cells grown at 25 and 37°C, further confirming that TLC1

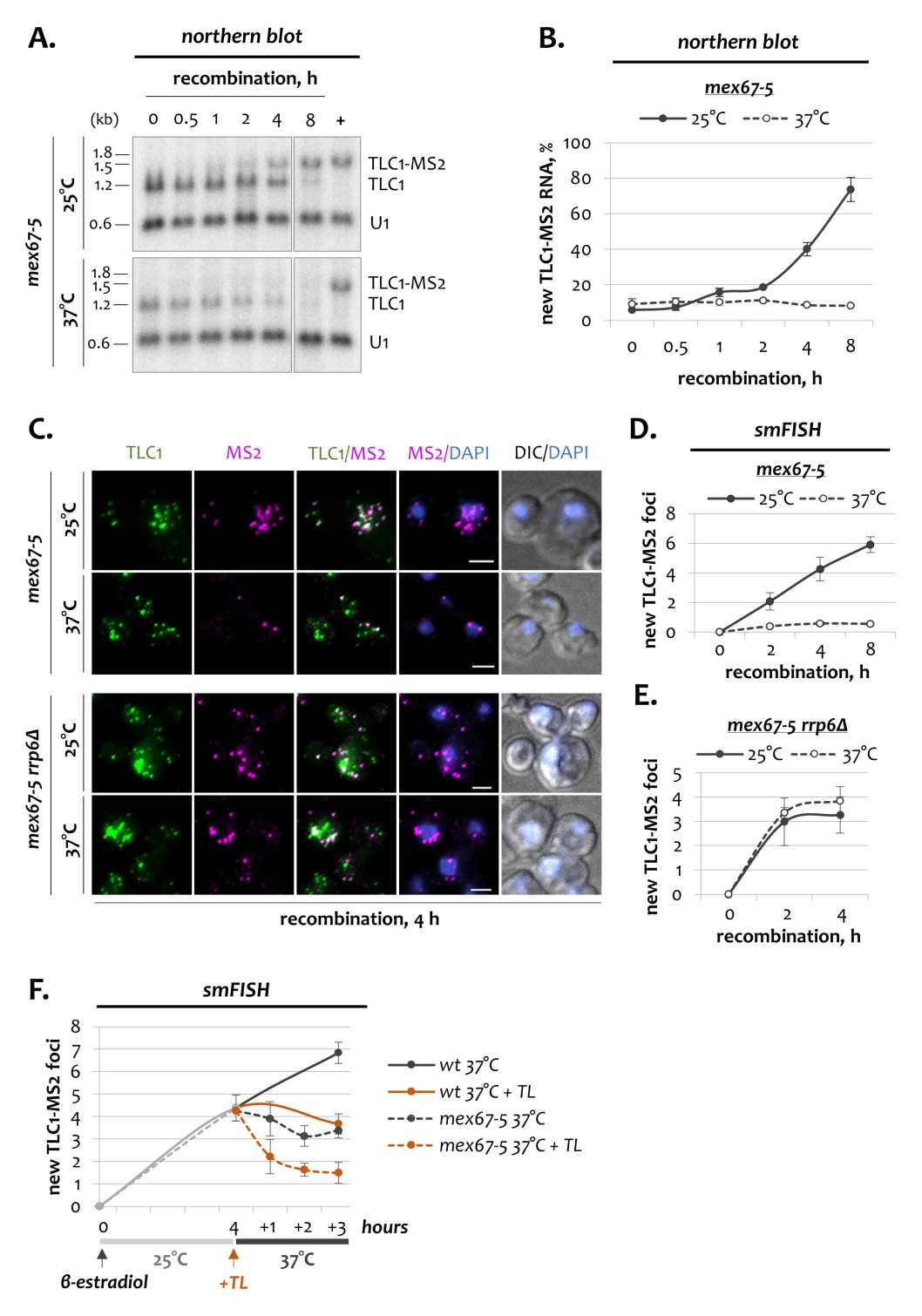

**Figure 4.** Dynamics of telomerase RNA fractions upon inactivation of Mex67. (**A**) Northern blot analysis of untagged TLC1 and tagged TLC1-MS2 RNA transcripts following induction of recombination in *mex67-5* cells grown at 25 and 37°C. Position of the TLC1 probe used to detect TLC1 species is shown in *Figure 1B*. U1 – loading control. '+' - positive recombined control. RNA sizes are indicated on the left (kb), 1.8 kb corresponds to 18S rRNA. (**B**) Quantification of the northern blots shown in A. Fractions relative to the positive recombined control (+) are shown. Error bars indicate SD, n = 3. (**C**) Two-color smFISH analysis of untagged TLC1 and tagged TLC1-MS2 molecules in *mex67-5* cells grown at 25 and 37°C after 4 hr of recombination. Scale bar, 2 μm. (**D, E**) Average
*Figure 4 continued on next page*

*Figure 4 continued*

number of new TLC1-MS2 foci detected by smFISH following recombination induction in *mex67-5* (D) and *mex67-5 rrp6Δ* (E) cells grown at 25 and 37°C. 50–100 cells from three independent experiments were scored. Error bars indicate SD. (F). Average number of new TLC1-MS2 foci detected by smFISH in *mex67-5* and wild type cells grown in the presence of thiolutin. After addition of β-estradiol, cells were grown for 4 hr at 25°C, followed by addition of thiolutin (TL) and temperature shift to 37°C. 50–100 cells from three (*mex67-5*) and two (*wt*) independent experiments were scored. Error bars indicate SD.

The online version of this article includes the following source data and figure supplement(s) for figure 4:

**Source data 1.** Numerical data used to generate *Figure 4*.
**Figure supplement 1.** TLC1 dynamics in *mex67-5* cells.
**Figure supplement 1—source data 1.** Numerical data used to generate *Figure 4—figure supplement 1*.
**Figure supplement 2.** TLC1 dynamics in *mex67-5 xpo1-1* cells.
**Figure supplement 2—source data 1.** Numerical data used to generate *Figure 4—figure supplement 2*.

---

transcription is not affected by inactivation of the Mex67 protein (*Figure 4—figure supplement 1E*). As a further test for the instability of the newly transcribed TLC1 RNA in *mex67-5* cells, we induced *TLC1-[MS2-IN]* recombination in *mex67-5 rrp6Δ* cells, in which the nuclear exosome complex, responsible for RNA quality control and decay, is inactive. Strikingly, deletion of *RRP6* completely suppressed the *mex67-5* phenotype, as in *mex67-5 rrp6Δ* cells grown at the restrictive temperature, new TLC1 RNA molecules accumulated to the same level as at the permissive conditions (*Figure 4C, E*; *Figure 4—figure supplement 1F,G*). These data directly show that in the absence of Mex67, new TLC1 transcripts undergo rapid degradation by the nuclear exosome and indicate that Mex67 is essential for stabilization of TLC1 RNA immediately after transcription and prior to export. At the same time, the stability of pre-existing and matured molecules does not seem to be affected by dysfunctional Mex67 (*Figure 4C*; *Figure 4—figure supplement 1B*). To confirm that Mex67 acts upstream of the Xpo1 exportin, we combined *mex67-5* and *xpo1-1* temperature-sensitive alleles in the same strain. In *mex67-5 xpo1-1* cells grown at the permissive temperature, newly synthesized TLC1 molecules accumulated with normal kinetics (*Figure 4—figure supplement 2A–C*). However, at 37°C, new TLC1 RNA was almost undetectable in a double mutant, thus completely recapitulating the phenotype of *mex67-5* cells. These data confirm that Mex67 indeed functions upstream of Xpo1, stabilizing newly transcribed TLC1 RNA prior to their cytoplasmic export.

While the vast majority of newly synthesized TLC1 RNA was unstable in the absence of Mex67, a few molecules still could be detected (*Figure 4C,D*). Notably, these remaining TLC1 foci almost exclusively accumulated at the nuclear periphery (*Figure 4C*; *Figure 4—figure supplement 1C*). This observation became even more striking in *mex67-5 rrp6Δ* double mutant cells, in which the vast majority of new molecules displayed this localization (*Figure 4C*; *Figure 4—figure supplement 1H*). Therefore, in addition to its essential role in stabilizing new TLC1 molecules, Mex67 acts as a prerequisite for the Xpo1-mediated export of TLC1 transcripts.

Collectively, the data thus demonstrate that Mex67 performs an entirely new and unexpected function in the biogenesis of telomerase RNA. Unlike in cells deficient for the Xpo1 exportin, which accumulate new telomerase RNA transcripts in the nucleus (*Figure 3A,B*), inactivation of Mex67 leads to an inability to produce stable and export-competent new TLC1 molecules (*Figure 4*).

## The Sm$_7$ complex is required for TLC1 stability after cytoplasmic export and for its re-import to the nucleus

According to our new data, Mex67 is essential for stability of newly synthesized telomerase RNA before its nuclear export (*Figure 4*). However, it is dispensable for stability of old molecules that have passed through the cytoplasm and returned to the nucleus. This suggests that after export to the cytoplasm, new TLC1 RNA acquires certain features that render it stable in the absence of a functional Mex67 protein. The Sm$_7$ complex is well known to stabilize telomerase RNA as well as most snRNAs by binding near the RNA 3'-end (*Seto et al., 1999*). Notably, recent data are consistent with the idea that snRNAs shuttle to the cytoplasm to associate with the Sm$_7$ proteins (*Becker et al., 2019*). Therefore, we hypothesized that telomerase RNA may also need to shuttle to the cytoplasm for the assembly with the Sm$_7$ complex and Sm$_7$-mediated stabilization.

To verify this idea, we took advantage of a mutated *tlc1-Sm2T* allele which disrupts $Sm_7$ binding to TLC1 and renders it unstable (*Seto et al., 1999*). We modified the inducible *TLC1-[MS2-IN]* locus such that before recombination, wild type TLC1 RNA proficient in $Sm_7$ binding is expressed. However, after the recombination switch, the new RNA molecules will harbor the tlc1-Sm2T mutation at their 3'-end (*Figure 5—figure supplement 1A*). Since only the tlc1-Sm2T RNAs have the MS2 tag, these molecules can be identified and tracked within the cell.

At the beginning of the time course, tlc1-Sm2T-MS2 molecules shuttle to the cytoplasm (*Figures 5A, B*, 1 h). However, to our surprise, these molecules never returned to the nucleus and remained in the cytoplasm (*Figures 5B*, 4 h). This suggests that the $Sm_7$ complex is required for re-import of new TLC1 transcripts to the nucleus. Although the kinetics of *TLC1-[Sm2T-MS2-IN]* recombination were normal (*Figure 5—figure supplement 1B*), the level of tagged tlc1-Sm2T-MS2 RNA remained low over the entire time course, as detected by northern blotting using the TLC1 or MS2 probes (*Figure 5C,D*). Consistently, the number of newly synthesized tlc1-Sm2T-MS2 foci was also lower than in wild type cells (*Figure 5—figure supplement 1C*). This indicates that new tlc1-Sm2T-MS2 transcripts are unstable, as expected (*Seto et al., 1999*). As a result of TLC1 loss, cells with the recombined *TLC1-[Sm2T-MS2-IN]* locus eventually undergo senescence and switch to a recombination-dependent telomere maintenance with the survivor telomere phenotype (*Figure 5—figure supplement 1D*). Since the remaining tlc1-Sm2T RNA accumulates in the cytoplasm (*Figure 5B*), we hypothesized that even in the absence of $Sm_7$ binding, newly transcribed TLC1 molecules are processed and exported from the nucleus normally, yet they become unstable and disappear after export to the cytoplasm. This scenario would predict that blocking the export of tlc1-Sm2T molecules to the cytoplasm should render them more stable. Indeed, in *tlc1-Sm2T xpo1-1* cells grown at the restrictive temperature, new tlc1-Sm2T RNA molecules accumulated in the nucleus and were more abundant than at the permissive conditions, when tlc1-Sm2T RNA shuttled to the cytoplasm (*Figure 5A,E–G*; *Figure 5—figure supplement 1E*). Therefore, our data show that newly synthesized telomerase RNA is stable in the nucleoplasm, even without the $Sm_7$ complex. The $Sm_7$ complex binds new TLC1 RNA only after its export to the cytoplasm, where this association becomes essential for TLC1 stability as well as for re-import of the molecules back into the nucleus.

## Discussion

As for any RNA species, the cellular pool of telomerase RNA encompasses a variety of molecules at different stages of biogenesis. Conventional imaging methods that rely on RNA-specific probes or tags visualize the total RNA pool and therefore yield information about the steady state RNA distribution. However, the diverse characteristics of various RNA sub-populations generally remain hidden in such assays. In this study, we overcame this limitation by combining fluorescent imaging with an inducible RNA tagging approach and applying it to the telomerase RNA TLC1. In this system, an inducible recombination event leads to a rearrangement of the *TLC1* genomic locus and the expression of TLC1 molecules tagged with MS2 repeats (*Figure 1*). Using a probe specific for the MS2 tag, we can selectively identify and analyze newly synthesized telomerase RNA transcripts within a heterogeneous TLC1 population and track their specific movements by smFISH (*Figure 2*). In addition, the dynamic behavior of TLC1 transcripts that remain untagged after recombination is complete, provides clues about the old or matured TLC1 molecules.

At steady state, 30–40% of telomerase RNA molecules localize in the cytoplasm (*Figures 2D*, 0 h) (*Gallardo et al., 2008*). However, the nature of the cytoplasmic TLC1 molecules remained enigmatic. According to the nucleo-cytoplasmic shuttling model, telomerase RNA gets exported to the cytoplasm for biogenesis purposes (*Ferrezuelo et al., 2002*; *Gallardo et al., 2008*). It was possible that TLC1 also accumulated in the cytoplasm at the end of its life cycle. Using the inducible TLC1 tagging approach, we show that in fact both models apply. Accordingly, we observed a temporary accumulation of newly synthesized TLC1 transcripts in the cytoplasm immediately after recombination induction, followed by their gradual return to the nucleus and re-establishment of a steady state RNA distribution (*Figure 2E*). Thus, our results provide direct evidence for the nucleo-cytoplasmic TLC1 shuttling model and narrow down this process to an early post-transcriptional event. Moreover, we show that old untagged TLC1 RNA accumulates in the cytoplasm by the end of the time course experiment (*Figures 2D*, 16 h), indicating a possible cytoplasmic decay of TLC1 molecules at the end of their life cycle. It still needs to be determined what triggers the exit of old transcripts from

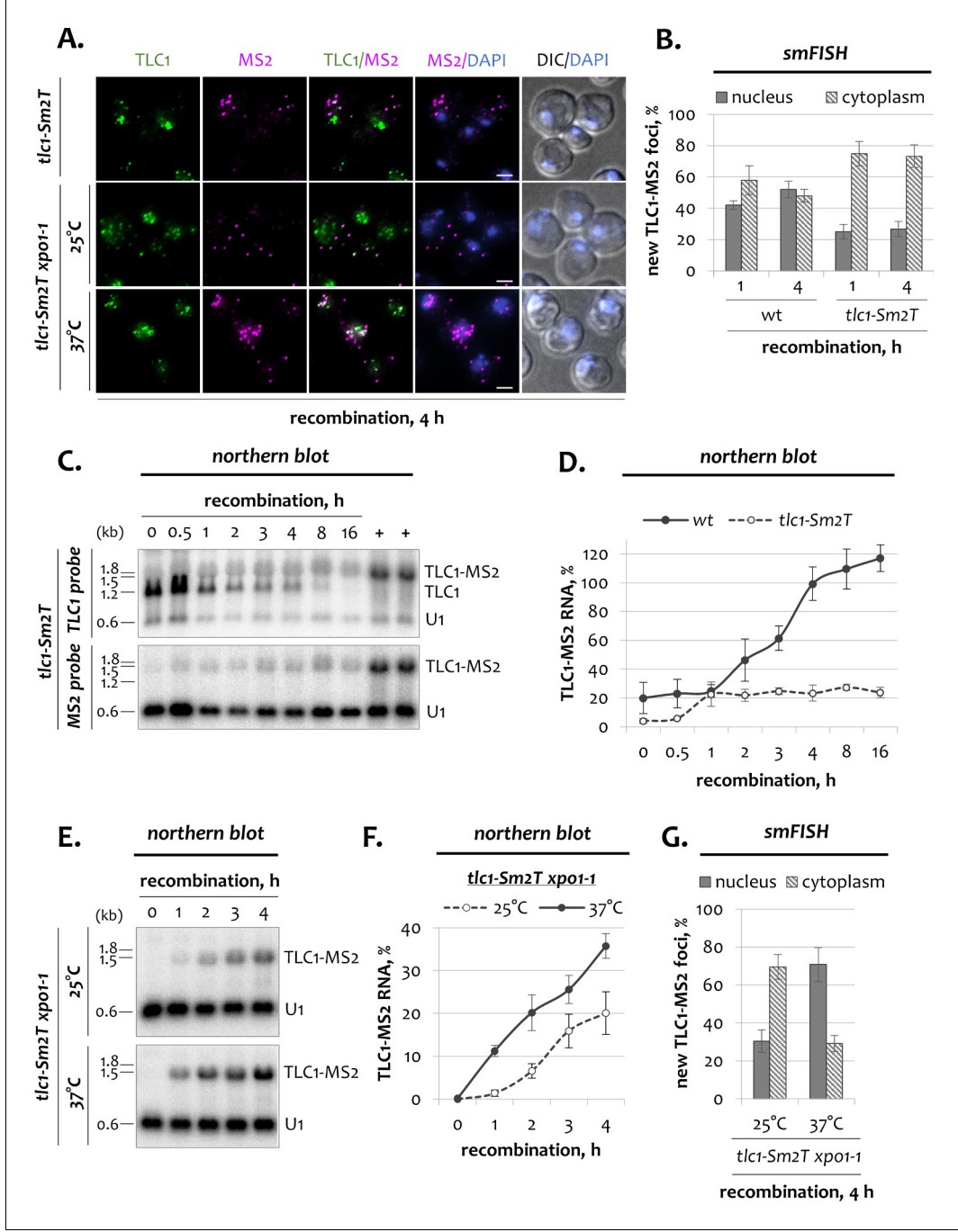

**Figure 5.** Dynamics of telomerase RNA fractions in *tlc1-Sm2T* mutants. (**A**) Two-color smFISH analysis of untagged TLC1 and tagged TLC1-MS2 molecules in *tlc1-Sm2T* and *tlc1-Sm2T xpo1-1* mutants after 4 hr of recombination. TLC1-Cy3 and MS2-Cy5 probes were used. Scale bar, 2 μm. (**B**) Relative nuclear and cytoplasmic fractions of tagged TLC1-MS2 foci detected by smFISH in wild type and *tlc1-Sm2T* cells after 1 hr and 4 hr of recombination. 50–100 cells from three independent experiments were scored. Error bars indicate SD. (**C**) Northern blot analysis of untagged TLC1 and tagged TLC1-MS2 RNA transcripts following induction of recombination in *tlc1-Sm2T* cells. The top blot was hybridized with the TLC1 probe depicted in *Figure 1B*. The bottom blot was hybridized with the MS2-specific probe. U1 – loading control. '+' - positive recombined controls. RNA sizes are indicated on the left (kb), 1.8 kb corresponds to 18S rRNA. (**D**) Quantification of the tagged TLC1-MS2 RNA from the MS2-specific northern blot shown in C. Fractions relative to the positive recombined controls (+) are shown. Error bars indicate SD, n = 2. Quantification of the tagged TLC1-MS2 RNA in wild type cells from *Figure 1E,F* is shown as a reference. (**E**) Northern blot analysis of tagged TLC1-MS2 RNA transcripts following induction of recombination in

*Figure 5 continued on next page*

*Figure 5 continued*

*tlc1-Sm2T xpo1-1* cells grown at 25 and 37˚C. The blot was hybridized with the MS2-specific probe. U1 – loading control. RNA sizes are indicated on the left (kb), 1.8 kb corresponds to 18S rRNA. (**F**) Quantification of the northern blot shown in (**E**). Fractions relative to the positive recombined controls are shown. Error bars indicate SD, n = 2. (**G**) Relative nuclear and cytoplasmic fractions of tagged TLC1-MS2 foci detected by smFISH in *tlc1-Sm2T xpo1-1* cells grown at 25 and 37˚C after 4 hr of recombination. 50–100 cells from two independent experiments were scored. Error bars indicate SD.

The online version of this article includes the following source data and figure supplement(s) for figure 5:

**Source data 1.** Numerical data used to generate *Figure 5*.
**Figure supplement 1.** Dynamics of TLC1 fractions in *tlc1-Sm2T* and *tlc1-Sm2T xpo1-1* cells.
**Figure supplement 1—source data 1.** Numerical data used to generate *Figure 5—figure supplement 1*.

the nucleus. Nevertheless, a significant number of old TLC1 molecules remained in the nucleus by the end of the time course (*Figures 2D*, 16 h). Hence, it is possible that telomerase RNA may also get degraded via a nuclear mechanism. In addition, we cannot exclude that TLC1 RNA undergoes additional rounds of shuttling at later stages of its life cycle. However, we favour the idea that untagged TLC1 foci that remain in the nucleus are the result of late recombination events that occurred in a subset of cells (*Figure 1C–G*). In this scenario, by the end of the time course, the majority of untagged TLC1 molecules are truly 'old' and localize in the cytoplasm for degradation, whereas a subset of untagged TLC1 is still functional and operates in the nucleus. Consistent with this idea, telomerase RNA has a very long-half life and can remain functional for much longer than 60 min, exceeding the length of one full cell cycle (*Chapon et al., 1997*; *Larose et al., 2007*).

The Xpo1 exportin and Kap122 importin were previously shown to affect the subcellular distribution of telomerase RNA (*Gallardo et al., 2008*). However, we aimed to understand whether these karyopherins are responsible for the nucleo-cytoplasmic shuttling of newly synthesized telomerase RNA transcripts (*Figure 3A–C*). Importantly, our results show that export of new telomerase RNA is entirely dependent on the Xpo1 exportin. Accordingly, in cells with dysfunctional Xpo1, newly transcribed TLC1 molecules accumulated exclusively in the nucleus, which would not be the case if an additional export receptor participated in telomerase RNA export (*Figure 3A,B*). On the other hand, Kap122 importin promotes the re-import of new TLC1 molecules to the nucleus, as following export in *kap122Δ* cells, most TLC1 molecules remained in the cytoplasm (*Figure 3A,C*). However, unlike TLC1 export which depends exclusively on Xpo1, nuclear re-import of new telomerase RNA molecules appears to be mediated by several importins. Accordingly, at later stages after the recombination initiation, the nuclear TLC1 fraction gradually increased in *kap122Δ* cells (*Figure 3C*). This alternative pathway could be dependent on Mtr10, which was previously shown to affect subcellular TLC1 localization (*Ferrezuelo et al., 2002*; *Gallardo et al., 2008*). Existence of a secondary Kap122-independent re-import mechanism for newly synthesized TLC1 transcripts is also supported by the fact that telomere maintenance is only mildly affected in the absence of Kap122 (*Figure 2—figure supplement 1D*). Interestingly, the maximal level of newly synthesized TLC1 RNA is decreased in *kap122Δ* cells (*Figure 3D,E*), suggesting that TLC1 that cannot re-enter the nucleus is degraded during the cytoplasmic phase. Therefore, a cooperation of several importins, including Kap122, facilitates a quick and efficient return of new TLC1 transcripts to the nucleus (*Figure 6*). Failure or delay in such re-import may lead to loss of telomerase RNA.

There is evidence that the karyopherin-unrelated export receptor Mex67 may cooperate with the Xpo1 exportin in the cytoplasmic export of several RNA species, including snRNAs, rRNAs and tRNAs (*Yao et al., 2007*; *Faza et al., 2012*; *Chatterjee et al., 2017*; *Becker et al., 2019*). However, the molecular mechanism of this Mex67 function is not well understood. Here, we decided to test whether Mex67 affects the nucleo-cytoplasmic shuttling of new telomerase RNA transcripts. Strikingly, newly synthesized TLC1 molecules almost completely disappear upon inactivation of Mex67 (*Figure 4A–D*). Using thiolutin-induced transcription block and RNA Pol II ChIP experiments, we ruled out the possibility that dysfunctional Mex67 causes the defect in TLC1 transcription (*Figure 4F*; *Figure 4—figure supplement 1E*). Instead, inactivation of Mex67 results in a nearly complete loss of new telomerase RNA after transcription – a phenotype that can be fully suppressed by removal of the Rrp6 exosome component (*Figure 4C,E*; *Figure 4—figure supplement 1F,G*). The few TLC1 foci that remained detectable upon inactivation of Mex67 almost exclusively localized at

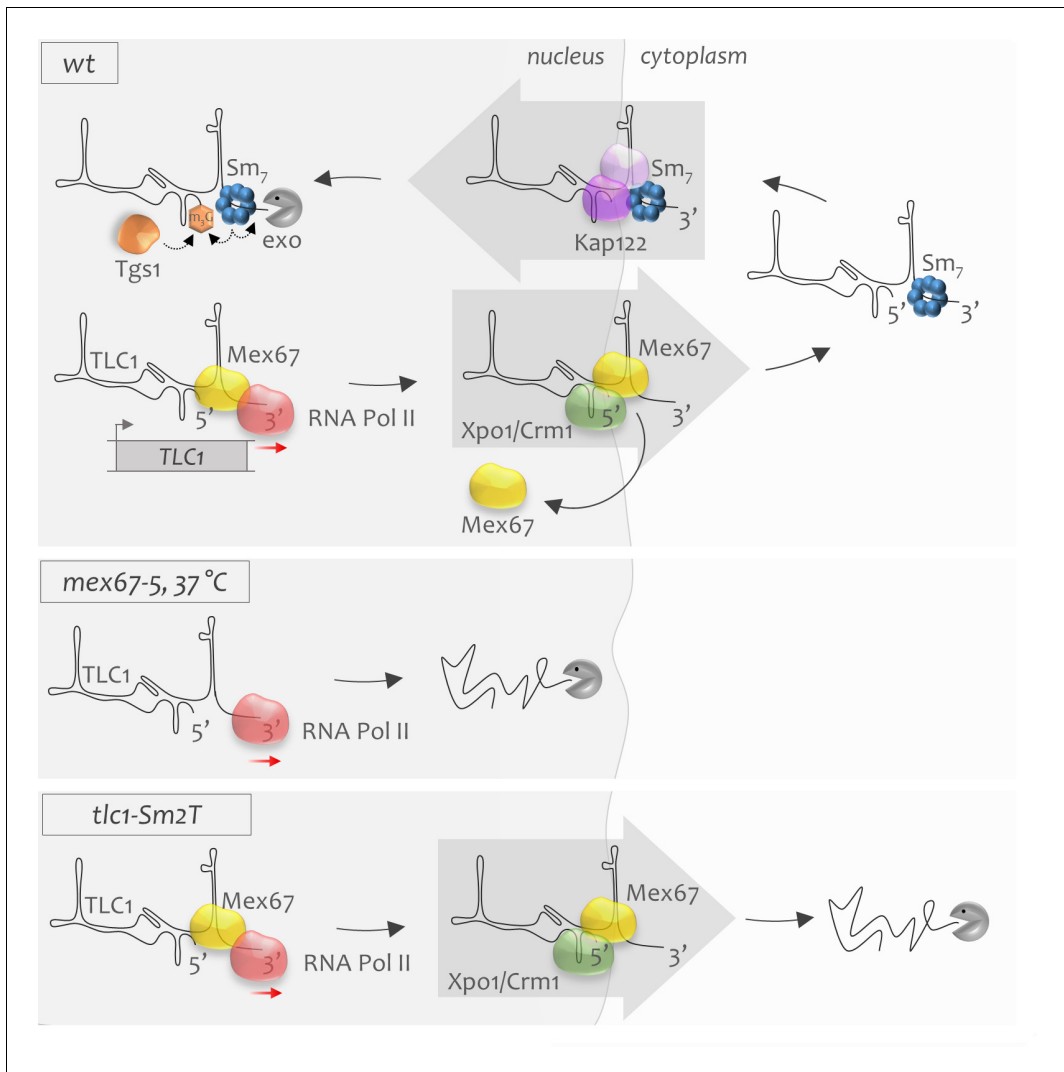

**Figure 6.** The model of early stages of telomerase RNA biogenesis. **WT:** Telomerase RNA is transcribed by RNA polymerase II (*red sphere*). Mex67 possibly associates co-transcriptionally with telomerase RNA, ensures its stability and protects the RNA 3'-end. Mex67 might also deliver TLC1 to the nuclear pores where it may associate as an adaptor with the Xpo1 exportin. Xpo1 mediates cytoplasmic export of new telomerase RNA transcripts. Mex67 will disembark from TLC1 in the cytoplasm and is recycled back to the nucleus. During the cytoplasmic stage, the heptameric Sm$_7$ complex binds the 3'-end of TLC1 which is required for stabilization of the RNA. Sm$_7$ binding may also promote TLC1 re-import to the nucleus. Nuclear re-import of TLC1 is mediated by Kap122 and other importins. Upon re-entry to the nucleus, TLC1 undergoes Sm$_7$-mediated 5'-TMG capping and 3'-end processing, which terminates the biogenesis cycle of telomerase RNA. *mex67-5, 37°C:* When Mex67 is dysfunctional, TLC1 is transcribed but is extremely unstable, which leads to rapid degradation of newly synthesized telomerase RNA transcripts in the nucleus. *tlc1-Sm2T:* Mutant TLC1 RNA that cannot associate with the Sm$_7$ complex is stable in the nucleus. However, after export to the cytoplasm, it becomes unstable and undergoes quick degradation.

the nuclear border (*Figure 4C*; *Figure 4—figure supplement 1C*), suggesting that in the absence of Mex67, new TLC1 transcripts quickly degrade in the nucleus and never make it to the cytoplasm. In addition, new TLC1 RNA was also unstable in a *mex67-5 xpo1-1* double mutant grown at the restrictive temperature (*Figure 4—figure supplement 2*), similar to the phenotype observed in *mex67-5* single mutants (*Figure 4A–D*). Altogether, these data show that Mex67 is essential for the nuclear stability of newly transcribed telomerase RNA prior to its Xpo1-mediated cytoplasmic export. On

the other hand, old TLC1 molecules, which remained untagged during the experiment, were not destabilized by inactivation of Mex67 (*Figure 4C*, *Figure 4—figure supplement 1B*).

An earlier study reported that in *mex67-5* cells grown at the restrictive temperature, telomerase RNA molecules are lost from the cytoplasm, which was interpreted as nuclear accumulation of TLC1 RNA (*Wu et al., 2014*). Our data here show that newly synthesized molecules largely contribute to the cytoplasmic TLC1 fraction and their disappearance in the absence of Mex67 indeed results in a decrease in the cytoplasmic TLC1 RNA. Notably, in this situation the nuclear TLC1 signal is composed of only matured or old RNA and there is no accumulation of new TLC1 molecules in the nucleus. Hence, the power of our inducible RNA tagging approach allows a much more targeted and detailed assessment of the dynamics of the complex TLC1 RNA population.

Similar to our results, depletion of Mex67 was shown to cause rapid nuclear degradation of all newly transcribed poly-A$^+$ mRNAs (*Tudek et al., 2018*). This untimely RNA decay was attributed to the role of Mex67 as an mRNA export receptor. Mechanistically, it was suggested that this effect involves Nab2 that binds to poly-A$^+$ tails and protects RNA transcripts from degradation (*Hector et al., 2002*; *Schmid et al., 2015*). The authors hypothesized that upon export block by Mex67 inactivation, Nab2 cannot be recycled from its substrate. Thus, accumulating poly-A$^+$ RNA tails will sequester the nuclear Nab2 pool away from newly formed RNA, rendering them unstable (*Tudek et al., 2018*). It is important to note that the vast majority of TLC1 (95%) is generated directly via the Nrd1/Nab3 non-coding RNA termination pathway (*Noël et al., 2012*). So far, Nab2 has not been implicated in this pathway, but we cannot exclude the possibility that other maturation factors specific to the non-coding RNA termination pathway could play a role similar to Nab2. Hence, the model suggested by Tudek et al. could also explain premature degradation of new TLC1 transcripts upon inactivation of Mex67 observed in our study (*Figure 4*). However, very much distinct from the above model, our results argue that Mex67 does not act as a classical TLC1 export receptor and an export block cannot explain the degradation of new TLC1 molecules. First, we show that export of new telomerase RNA molecules to the cytoplasm is entirely dependent on Xpo1 (*Figure 3A,B*). If Mex67 operated in parallel with Xpo1 in TLC1 export, at least a fraction of new telomerase RNA molecules would be expected to eventually reach the cytoplasm in *xpo1-1* cells grown at the restrictive temperature, which we did not observe (*Figure 3A,B*). Second and more importantly, although inactivation of Xpo1 completely blocks TLC1 in the nucleus, these molecules are perfectly stable and their number even increases beyond the level observed in wild type cells (*Figure 3D,E*). This means that at least for telomerase RNA, an export block *per se* does not lead to nuclear RNA decay. In other words, while in cells with dysfunctional Mex67 newly transcribed TLC1 RNA is completely lost, it is retained and accumulates in the nucleus in *xpo1-1* mutants grown at the restrictive temperature (*Figure 4A–D*). This suggests that Mex67 acts upstream of Xpo1 and plays an unexpected role in stabilization of newly transcribed TLC1 molecules before their cytoplasmic export (*Figure 6*).

Hence, our results indicate a novel role of Mex67 in stabilization of telomerase RNA which to our knowledge is the first time that Mex67 is implicated in a function distinct from RNA export. Mex67 is known to bind at least to some of its cargoes co-transcriptionally (*Dieppois et al., 2006*; *Wende et al., 2019*). It has been proposed that such early association with its RNA substrates may assist in proper 3'-end RNA maturation and protection (*Forrester et al., 1992*; *Hammell et al., 2002*; *Qu et al., 2009*). Failure to execute this function may lead to elimination of faulty transcripts as was previously reported for other 3'-end processing mutants (*Minvielle-Sebastia et al., 1991*; *Forrester et al., 1992*; *Hammell et al., 2002*). There was some evidence that Mex67 may be able to associate with telomerase RNA (*Wu et al., 2014*), and our results here firmly establish that Mex67 protects the 3'-end of TLC1 from nuclear degradation by Rrp6 (*Figure 4C,E*; *Figure 4—figure supplement 1F–H*; *Figure 6*). Moreover, since Mex67 can bind to both RNA and nuclear pore components, it may be responsible for a quick delivery of new transcripts to the nuclear pore where they are stabilized and exported by Xpo1. The latter is consistent with our finding that the few newly synthesized molecules that are still visible in the *mex67-5* mutant localize at the nuclear border (*Figure 4C*). Given that Mex67 has been implicated in the life cycle of diverse RNA classes, we speculate that Mex67 may perform a stabilization function for the poly-A$^+$ RNA species as it does for telomerase RNA.

Since old TLC1 molecules underwent at least one round of nucleo-cytoplasmic shuttling, we reasoned that during this shuttling, telomerase RNA must have acquired a property that renders it insensitive to the loss of Mex67. Hence, we asked which telomerase subunits could confer TLC1

stability after export to the cytoplasm. Using a modified inducible TLC1 tagging system in which the new TLC1 RNA carries a tlc1-Sm2T mutation, we demonstrate that TLC1 RNA deficient in $Sm_7$ binding accumulates in the cytoplasm (*Figure 5*; *Figure 5—figure supplement 1*). Given the nuclear instability of the wild type RNA in the absence of Mex67 (see above), we conclude that the $Sm_7$ complex associates with the TLC1 RNA in the cytoplasm and that this association in turn stabilizes the RNA. Recently a cytoplasmic association with the $Sm_7$ complex has also been suggested for snRNAs (*Becker et al., 2019*). Similarly, nuclear export and cytoplasmic $Sm_7$ binding is also a characteristic of mammalian snRNAs (*Fischer et al., 2011*).

The data reported here challenge the current view of telomerase biogenesis by placing the $Sm_7$ binding step after export of telomerase RNA to the cytoplasm (*Figure 6*). The new model incorporates the idea that in the nucleus, newly synthesized poly-A⁻ telomerase RNA (*Noël et al., 2012*) first requires stabilization by Mex67, which may also usher TLC1 to nuclear pores. Whether the Mex67 association occurs co-transcriptionally or in coordination with the 3'-end formation or even after transcript release remains to be determined. Failure to associate with Mex67 results in immediate degradation of new TLC1 transcripts via the nuclear exosome. Next, the RNA is transported to the cytoplasm which absolutely requires Xpo1. This karyopherin does not bind RNA directly and we still do not know what the TLC1 adaptor for Xpo1 is. In this regard, our data are consistent with the possibility that Mex67 itself plays this role in addition to its stabilizing function in the nucleus (*Figure 6*). If Mex67 indeed acts as an adaptor between TLC1 RNA and Xpo1 exportin, it would likely translocate through the nuclear membrane with the Xpo1-TLC1 complex. Once in the cytoplasm, the TLC1 RNA must rapidly associate with the $Sm_7$ complex, given that Mex67 will likely disembark from the RNA at the nuclear pore exit and get recycled back into the nucleus (*Stewart, 2007*). At this stage then, the $Sm_7$ complex is required for TLC1 RNA stabilization, and in turn, TLC1 stability allows for other telomerase subunits to associate with the maturing RNP. If this step is compromised, TLC1 molecules undergo degradation in the cytoplasm. Whether the cytoplasmic exosome, the Xrn1 nuclease or other machineries perform this function remains to be determined. Eventually, the telomerase RNP is re-imported to the nucleus primarily via Kap122, but alternative and less efficient import pathways do exist (*Figure 3C*). Finally, our model suggests that the final 5'-TMG capping and 3'-end processing of telomerase RNA occurs in the nucleus after its re-import and is dependent on the presence of the $Sm_7$-complex, possibly analogous to snRNA maturation (*Becker et al., 2019*). This final step may thus occur on a relatively mature telomerase RNP, which is entirely feasible given a predicted close proximity of the 5'- and 3'-ends of telomerase RNA (*Cech, 2004*; *Dandjinou et al., 2004*). It is also consistent with the fact that active telomerase RNPs only contain 5'-TMG capped telomerase RNAs (*Bosoy et al., 2003*). Maturation of telomerase RNA may also serve as a final stabilization step for TLC1 RNA and render it insensitive to loss of Mex67, as observed above (*Figure 4C*).

Our study thus elucidates many of the early events of telomerase RNA biogenesis. However, later stages of TLC1 life cycle remain enigmatic, e.g., does telomerase RNA continue to cycle between the nucleus and the cytoplasm or whether these and final shuttling rounds rely on the same machinery as the initial export and re-import. Although our data show that Mex67 is dispensable for TLC1 stability after the first round of shuttling (*Figure 4*), it might still act as a TLC1-Xpo1 adaptor for the subsequent rounds of export. Alternatively, TLC1 might use a different adaptor for later shuttling steps or even rely on a different karyopherin (such as Los1 or Msn5).

As in yeast, the generation and stabilization of human telomerase RNA relies on the association with several proteins, including dyskerin, NHP2 and NOP10 as well as factors involved in the noncoding RNA maturation pathway such as PARN (*Hoareau-Aveilla et al., 2006*; *Fu and Collins, 2007*; *Walne et al., 2007*; *Moon et al., 2015*; *Nguyen et al., 2015*). Mutations in PARN, dyskerin or NOP10 cause severe genetic disorders, emphasizing the functional significance of these telomerase subunits and RNA maturation mechanisms (*Vulliamy et al., 2006*; *Walne et al., 2007*). Furthermore, the loss of telomerase RNA due to PARN mutations is amenable to reversion by small molecules that thus may alleviate the severe associated disease (*Brenner and Nandakumar, 2020*). However, certain details of human telomerase biogenesis remain enigmatic. For example, how and when do dyskerin and NOP10 associate with telomerase RNA and is association with these subunits needed for RNA processing? Inducible RNA tagging tools as described in this study should greatly aid in teasing out these maturation steps and may provide missing links in human telomerase biogenesis.

# Materials and methods

## Key resources table

| Reagent type (species) or resource | Designation | Source or reference | Identifiers | Additional information |
|---|---|---|---|---|
| Strain, strain background (*S. cerevisiae*) | All strains (W303) | *Supplementary file 1* | | |
| Gene (*S. cerevisiae*) | TLC1 | *Saccharomyces* Genome Database | TLC1; SGD:S000006657 | |
| Gene (*S. cerevisiae*) | tlc1-Sm2T | doi:10.1038/43694 | | |
| Gene (*S. cerevisiae*) | XPO1, CRM1 | *Saccharomyces* Genome Database | YGR218W; SGD:S000003450 | |
| Gene (*S. cerevisiae*) | xpo1-1 | gift from Pascal Chartrand doi:10.1016/s0092-8674(00)80370–0 | | temperature-sensitive allele |
| Gene (*S. cerevisiae*) | MEX67 | *Saccharomyces* Genome Database | YPL169C; SGD:S000006090 | |
| Gene (*S. cerevisiae*) | mex67-5 | gift from Marlene Oeffinger doi:10.1093/emboj /16.11.3256 doi:10.1038/nbt.1832 | | temperature-sensitive allele |
| Gene (*S. cerevisiae*) | KAP122 | *Saccharomyces* Genome Database | YGL016W; SGD:S000002984 | |
| Gene (*S. cerevisiae*) | RRP6 | *Saccharomyces* Genome Database | YOR001W; SGD:S000005527 | |
| Chemical compound, drug | β-estradiol | Sigma | Cat#: E8875 | Recombination induction, 12.5 µM final |
| Chemical compound, drug | ClonNAT, nourseothricin sulfate | BioShop | Cat#: NUR001 | Selection, 100 µg/ml final |
| Chemical compound, drug | Thiolutin | Abcam | Cat#: ab143556 | Transcription block, 3 µg/ml final |
| Chemical compound, drug | Para-formaldehyde solution, 32% | Electron Microscopy Sciences | Cat#: 15714 | FISH, 4% final |
| Chemical compound, drug | Ribonucleoside vanadyl complex, 200 mM | New England Biolabs | Cat#: S1402S | FISH, 20 mM final |
| Chemical compound, drug | Formamide | Sigma | Cat#: F9037 | FISH, 40% final |
| Chemical compound, drug | Salmon sperm DNA | Invitrogen | Cat#: 15632–011 | FISH, 0.4 mg/ml final |
| Chemical compound, drug | tRNA from *E. coli* | Roche | Cat#: 10109541001 | FISH, 0.4 mg/ml final |
| Chemical compound, drug | Poly-L-lysine, 0.01% | Sigma | Cat#: P4707 | |
| Chemical compound, drug | DAPI | Sigma | Cat#: D9542 | FISH, 0.5 µg/ml final |
| Chemical compound, drug | *p*-phenylenediamine | Sigma | Cat#: P6001 | |
| Commercial assay or kit | Cy3 Mono-Reactive dye | Cytiva | Cat#: PA23001 | |
| Commercial assay or kit | Cy5 Mono-Reactive dye | Cytiva | Cat#: PA25001 | |
| Commercial assay or kit | ThermoScientific Pierce Protein A/G Magnetic Beads | ThermoScientific | Cat#: 88802 | ChIP, 50 µl per sample |

*Continued on next page*

*Continued*

| Reagent type (species) or resource | Designation | Source or reference | Identifiers | Additional information |
|---|---|---|---|---|
| Antibody | Anti-RNA Polymerase II RPBI (mouse monoclonal antibody, clone 8WG16) | Cedarlane | Cat#: 664906 | ChIP (5 µl per sample) |
| Peptide, recombinant protein | XmnI (restriction enzyme) | New England Biolabs | Cat#: R0194L | |
| Peptide, recombinant protein | AleI-v2 (restriction enzyme) | New England Biolabs | Cat#: R0685L | |
| Peptide, recombinant protein | XhoI (restriction enzyme) | New England Biolabs | Cat#: R0146L | |
| Peptide, recombinant protein | Lyticase | Sigma | Cat#: L2524 | FISH, 5 U/ml final |
| Recombinant DNA reagent | All plasmids | *Supplementary file 2* | | |
| Sequence-based reagent | All primers | *Supplementary file 3* | | |
| Software, algorithm | SnapGene 2.1 | http://www.snapgene.com/ | RRID:SCR_015052 | |
| Software, algorithm | ImageQuant TL 8.2 (Cytiva) | https://www.cytivalifesciences.com/en/is/shop/protein-analysis/molecular-imaging-for-proteins/imaging-software/imagequant-tl-8-2-image-analysis-software-p-09518 | RRID:SCR_018374 | |
| Software, algorithm | GraphPad Prism 8 | http://www.graphpad.com/ | RRID:SCR_002798 | |
| Software, algorithm | ZEN 3.1 2019 (Zeiss) | https://www.zeiss.com/microscopy/int/products/microscope-software/zen.html | RRID:SCR_013672 | |

## Strains and plasmids

Strains used in this study are derivatives of W303. All strains are listed in *Supplementary file 1*. To express the inducible tagging system, the *TLC1-[MS2-IN]* cassette was integrated in the *TLC1* genomic locus. The *TLC1-[MS2-IN]* cassette containing the *mid-TLC1* sequence, *loxP* site, *TLC1* 3'-end, *natMX4* cassette, *ADH1* terminator, *loxP* site, *10xMS2* repeats followed by a duplicated *TLC1-3'* end, was amplified by PCR from pYV132 using qTLC1-F and TLC1 AvaI-R primers (plasmids and primers used in the study are listed in *Supplementary file 2* and *Supplementary file 3*, respectively). In the modified *TLC1-[Sm2T-MS2-IN]* tagging system, the sequence encoding the Sm$_7$-binding site (ATTTTTGG) in the duplicated *TLC1* 3'-end downstream of the *10xMS2* array was mutated into ATTGG, resulting in the expression of the *tlc1-Sm2T* mutant RNA upon recombination. To express such system, the *TLC1-[Sm2T-MS2-IN]* cassette was amplified from pYV142 using qTLC1-F and TLC1 AvaI-R primers. To express the Cre-EBD recombinase in wild type, *kap122* and *tlc1-Sm2T* backgrounds, pTW40 was linearized with NdeI and integrated at the *his3* locus. To express Cre-EBD in *xpo1-1*, *xpo1-1 tlc1-Sm2T*, *mex67-5*, *mex67-5 rrp6Δ* and *mex67-5 xpo1-1* mutants, pYV144 was linearized with NdeI and integrated at the *ura3* locus. YV359, YV360 recombined controls are clonNAT-sensitive derivatives of YV349 and YV352. *kap122Δ* and *rrp6Δ* mutants were constructed by a one step replacement of the *KAP122* and *RRP6* genes, respectively, with the *kanMX4* cassette, amplified from pRS400 using YV395_KAP122_F1 and YV396_KAP122_R1 primers for the deletion of *KAP122*, and *rrp6Δ_for* and *rrp6Δ_rev* primers for the deletion of *RRP6*. The *mex67-5 xpo1-1* double mutant strain was constructed by a one step replacement of the *MEX67* gene in YV365, YV366 with the *mex67-5-kanMX6* cassette, amplified from the Y11429 strain (gift from M. Oeffinger lab) using YV518_MEX67-237 and YV519_MEX67+278 primers.

pEB36 was constructed via multiple cloning steps. Sequence containing the fragment of the *TLC1* gene (665–1145 bp), the *loxP* site, the *TLC1* 3'-end (1146–1301 bp) followed by 148 bp of the

downstream sequence, the *natMX4* cassette (consisting of the *TEF* promoter, the *NAT^R* gene and *TEF* terminator), the *loxP* site, *10xMS2* repeats and the duplicated *TLC1* 3'-end (1146–1301 bp) followed by 1156 bp of the downstream sequence was cloned in pRS306 between BamHI and EcoRI restriction sites. pYV132 was constructed via multiple cloning steps, resulting in addition of the *ADH1* terminator sequence downstream of the *TEF* terminator in pEB36. pYV142 was constructed via multiple cloning steps, which led to the replacement of the $Sm_7$-binding site (ATTTTTGG) in the duplicated *TLC1* 3'-end of pYV132 with ATTGG sequence. pYV144 was constructed by replacing AatII - SacII *HIS3*-containing fragment of pTW40 with AatII - SacII *URA3*-containing fragment of pRS306. All plasmids constructed in this study were verified by sequencing.

## Yeast culture

For inducible *TLC1-[MS2-IN]* tagging experiments in wild-type cells (YV349, YV352; *Figures 1* and *2*), *kap122* mutants (YV355, YV356; *Figure 3*) and *tlc1-Sm2T* mutants (YV390, YV391; *Figure 5*), cells were pre-cultured overnight in SC-HIS media supplemented with clonNAT (100 µg/ml) and grown at 30°C in SC complete medium without clonNAT from OD 0.1 to 0.4. Recombination was induced with 12.5 µM β-estradiol (Sigma) and monitored for the duration of the time course. For the experiments at the restrictive temperature (*Figure 3—figure supplement 1D,G*), wild type cells (YV349, YV352) were shifted to 37°C 1 hr before addition of β-estradiol and grown for 8 hr at 37°C. For the experiments with the temperature-sensitive mutants *xpo1-1* (YV365, YV366 [*Figure 3*]; YV403, YV404 [*Figure 5A–G*]); *mex67-5* (YV385, YV386; YV399, YV400 [*Figure 4A–E*]) and *mex67-5 xpo1-1* (YV417, YV418 [*Figure 4—figure supplement 2*]), cells were pre-cultured overnight at 25°C in SC-HIS-URA (*xpo1-1 and mex67-5 xpo1-1*) and SC-LEU-URA (*mex67-5*) supplemented with clonNAT (100 µg/ml) and grown at 25°C in SC-HIS (*xpo1-1 and mex67-5 xpo1-1*) and SC-LEU (*mex67-5*) without clonNAT from OD 0.1 to 0.3. Cells were divided in two cultures and grown at 25 or 37°C for 1 hr. Recombination was induced with 12.5 µM β-estradiol and monitored for the indicated period of time at 25 or 37°C. For the experiment described in *Figure 4F*, wild type (YV349, YV352) and *mex67-5* cells (YV385, YV386) were pre-cultured overnight at 25°C in SC-HIS (wild type) and SC-LEU-URA (*mex67-5*) supplemented with clonNAT (100 µg/ml) and grown at 25°C in SC complete (wild type) and SC-LEU (*mex67-5*) media without clonNAT from OD 0.1 to 0.4. After addition of β-estradiol (12.5 µM), cells were grown at 25°C for 4 hr. Thiolutin (3 µg/ml, Abcam) was added to half of the cultures for 15 min. Thiolutin-treated and -untreated cultures were split and grown at 25 or 37°C for 3 hr. In all experiments, cell aliquots were collected for DNA, RNA and FISH analysis at regular time points. OD of cell cultures was maintained below 0.8 throughout the length of experiments.

To isolate recombined control strain (YV359, YV360), YV349 or YV352 cells were grown in YEPD media without clonNAT, treated with 12.5 µM β-estradiol overnight, plated on YEPD media for single colonies and replica-plated on YEPD + clonNAT (100 µg/ml) plates. clonNAT-sensitive clones were isolated.

## Southern blotting

Genomic DNA was extracted using phenol-chloroform (*Sambrook et al., 1989*). To monitor the *TLC1-[MS2-IN]* recombination, DNA was digested with XmnI and AleI-v2 (NEB), resolved on a 0.8% agarose-1x TBE gel, transferred on a Hybond-XL nylon membrane (Cytiva) and hybridized to $^{32}$P-labeled TLC1 and CEN4-1.5 kb probes. Probes were obtained by PCR amplification (see *Supplementary file 3* for primer sequences), followed by random priming labeling procedure (*Feinberg and Vogelstein, 1983*). For telomere length analysis, DNA was digested with XhoI (NEB) and subjected to agarose gel electrophoresis and transfer as described above. Blots were hybridized to a $^{32}$P-labeled pCT300 probe (300 bp fragment containing 280 bp of telomeric repeats derived from pYLPV [*Wellinger et al., 1993*]) and PCR-amplified $^{32}$P-labeled CEN4-1.6 kb probe. Blots were visualized using Typhoon FLA9000 (Cytiva) and quantified via Image Quant TL software (Cytiva).

To quantify the fraction of the unrecombined *TLC1-[MS2-IN]* DNA, the ratio between the fragment 'a' intensity (*Figure 1A*) at the time point x and the time point 0 (100% unrecombined locus) was calculated. To quantify the recombined *TLC1-[MS2-IN]* DNA fraction, the ratio between the fragment 'b' intensity at the time point x and in the recombined control strain (YV359, YV360) (100% recombined locus) was calculated. Intensity of fragments 'a' and 'b' was normalized to the *CEN4* loading control.

## Northern blotting

RNA extraction and northern blotting were performed as described previously (*Laterreur et al., 2018*). In brief, RNA was extracted 2x with phenol/chloroform/isoamyl alcohol (25:24:1) and 1x with chloroform/isoamyl alcohol (24:1). RNA was mixed with NaOAc (150 mM), glycogen (50 µg) and precipitated in 2x volume of 100% ethanol at −20°C. 10 µg of RNA was mixed with 1x MOPS (pH 7), 3.7% formaldehyde, 45% formamide and 1x RNA dye. Samples were migrated on a 1.1% agarose-1x MOPS (pH 7)–2% formaldehyde gel and transferred on a Hybond N+ membrane (Cytiva). After UV-crosslinking, the membrane was hybridized to $^{32}$P-labeled TLC1 or non-aminoallyl MS2 probe. The membrane was simultaneously hybridized to the U1 probe used to detect the U1 RNA as a loading control. The TLC1 probe was obtained by PCR amplification (see *Supplementary file 3* for primer sequences), followed by random priming labeling procedure (*Feinberg and Vogelstein, 1983*). An oligonucleotide non-aminoallyl MS2 and U1 probes were obtained by 5′-end labeling (*Sambrook et al., 1989*). Blots were visualized using Typhoon FLA9000 (Cytiva) and quantified via Image Quant TL software (Cytiva).

To quantify the fraction of untagged TLC1 RNA, the ratio between the intensities of the TLC1 band at the time point x and the time point 0 (100% untagged TLC1) was calculated. To quantify the fraction of tagged TLC1-MS2 fraction, the ratio between the intensities of the TLC1-MS2 band at the time point x and in the recombined control strain (YV359, YV360) (100% tagged TLC1-MS2) was calculated. Intensity of the TLC1 and TLC1-MS2 bands was normalized to the U1 loading control.

## RT-qPCR and ddPCR

RNA integrity was assessed with the Agilent 2100 Bioanalyzer (Agilent Technologies). Reverse transcription was performed on 2 µg total RNA using Transcriptor reverse transcriptase, random hexamers, dNTPs (Roche Diagnostics) and 10 units of RNAseOUT (Invitrogen) in a total volume of 10 µl following the manufacturer's protocol. For ddPCR setup part, qPCR reactions were performed in 10 µl in 384 well plates on a CFX-384 thermocycler (BioRad) with 5 µl of 2X PerfeCTa SYBR Green Supermix (Quantabio), 10 ng of cDNA and 200 nM primer pair solutions. The following cycling conditions were used: 3 min at 95°C; 50 cycles: 15 s at 95°C, 30 s at 60°C, 30 s at 72°C. Primer design and validation were evaluated as described elsewhere (*Brosseau et al., 2010*).

Droplet Digital PCR (ddPCR) reactions included 10 µl of 2X QX200 ddPCR EvaGreen Supermix (Bio-Rad), 240 ng of cDNA and 200 nM primer pair solutions in a 20 µl total volume. Each reaction mix was converted into droplets with the QX200 droplet generator (Bio-Rad). Droplet-partitioned samples were then transferred to a 96-well plate, sealed and cycled in a C1000 deep well Thermocycler (Bio-Rad) under the following cycling protocol: 95°C for 5 min, 50 cycles: 95°C for 30 s, 60°C for 1 min, 72°C for 30 s; post-cycling steps: 4°C for 5 min and 90°C for 5 min. RNA concentration (copies/ul) was determined using the QX200 reader (Bio-Rad) (*Taylor et al., 2015*). To quantify the fraction of old untagged TLC1 molecules, the ratio between untagged TLC1 RNA concentration (determined by primers TLC1.AB.qsc.F3 and TLC1.A.lna.qsc.R4; Supplemetary file 3) and total TLC1 concentration (primers TLC1.AB.qsc.F1 and TLC1.AB.qsc.R1) was calculated. To quantify the fraction of new tagged TLC1-MS2 molecules, the ratio between tagged TLC1-MS2 RNA concentration (determined by primers TLC1.AB.qsc.F3 and TLC1.B.qsc.R1) and total TLC1 concentration (primers TLC1.AB.qsc.F1 and TLC1.AB.qsc.R1) was calculated.

## Fluorescence *in situ* hybridization (FISH)

Cell fixation and preparation of spheroplasts was performed as previously described (*Gallardo and Chartrand, 2011*). Hybridization with the probes was performed as in *Gallardo and Chartrand, 2011* with slight modifications. Spheroplasts were washed with 2x SSC (2 × 5 min) and with 2x SSC containing 40% formamide (1 × 5 min) at room temperature. Centrifugation between washes was done at 1500 g for 3 min. Cells were resuspended in 2x SSC containing 40% formamide and incubated for 1 hr at 37°C. A mix of five TLC1-specific probes (*Supplementary file 3*) were conjugated with the Cy3 fluorophore (Cytiva), and the MS2 probe was conjugated with the Cy5 fluorophore according to the manufacturer's instructions. 10 ng of TLC1-Cy3 probe mix and 10 ng of the MS2-Cy5 probe were mixed with 4 µl of competitor DNA (10 mg/ml solution of 1:1 sonicated salmon sperm DNA (Invitrogen) / *E. coli* tRNA (Roche)), 50 µl of solution F (80% formamide and 10 mM NaHPO$_4$, pH 7.5). The probe solution was heated at 95°C for 5 min and mixed with 50 µl of solution

H (4x SSC, 20 mM ribonucleoside vanadyl complex (NEB), BSA 4 µg/ml). Spheroplasts were centrifuged at 1500 g for 3 min and resuspended in the probe mix. Hybridization was performed overnight at 37°C. After hybridization, cells were washed in 2x SSC 40% formamide for 6 min, 2x SSC 0.1% Triton X-100 for 6 min and 1x PBS for 6 min at room temperature. Cells were incubated in DAPI (Sigma) 0.5 µg/ml for 3 min and washed in 1x PBS for 3 min. Cells were resuspended in a small volume of 1x PBS and immobilized on coverslips coated with 0.01% poly-lysine (Sigma) for 30 min. Unattached cells were washed away from the coverslips with 70% ethanol and sequentially incubated in 70% ethanol, 2x SSC, 1x PBS and 100% ethanol for 3 s. After drying, coverslips were mounted on the microscope slides with the mounting medium (86% glycerol, 1 mg/ml *p*-phenylenediamine (Sigma)−1x PBS).

Image acquisition was performed using Zeiss mRm Axiocam mounted on an Axio Observer Z1 inverse microscope. To illuminate TLC1-Cy3, X-cite Arc lamp and 43HE filter (Zeiss) were used. For MS2-Cy5 channel, Zeiss Colibri LED 625 nm was used in combination with 77HE filter. DAPI was visualized with Zeiss Colibri LED 365 nm 62HE filter. Exposure times were 3 s for Cy3 and Cy5, and 500 ms for DAPI. Single plane images spanning 3 µm with steps of 200 nm were projected in a z-stack to facilitate data analysis. Image analysis and constrained iterative deconvolution was performed using ZEN 3.1 2019 software (Zeiss). Foci counting was performed manually. Foci with the colocalized TLC1- and MS2-specific signal were counted as new TLC1 molecules. Foci with only the TLC1-specific signal were counted as old TLC1 molecules. Relative TLC1 fractions (%) were calculated as a ratio of new or old molecules to the total number of TLC1 RNA. Relative nuclear or cytoplasmic TLC1 fractions (%) were calculated as a ratio of old or new molecules localized inside or outside the nucleus (as determined by DAPI staining) to the total number of old or new molecules detected in a single cell. For each experiment (time point), 50–100 single cells from two or three independent experiments were scored.

## Chromatin immunoprecipitation (ChIP)

200 ml of culture was pre-grown overnight at 25°C in SC-LEU-URA supplemented with clonNAT (100 µg/ml), followed by incubation at 25°C SC-LEU without clonNAT from OD 0.1 to 0.3. Cells were divided in two cultures (100 ml each) and grown at 25 or 37°C for 1 hr. Recombination was induced by addition of 12.5 µM β-estradiol (Sigma) for 2 hr at 25 or 37°C. ChIP was performed as described in *Pasquier and Wellinger, 2020* using 3 µl of RNA Polymerase II mouse monoclonal antibody (8WG16, Cedarlane) and 50 µl equilibrated ThermoScientific Pierce Protein A/G Magnetic Beads (ThermoScientific). Immunoprecipitated and input DNAs were subjected to qPCR at the *TLC1* locus using TLC1_G_for1 and TLC1_G_rev1 primers (*Supplementary file 3*). U1 RT F2 and U1 RT R2 primers were used for the qPCR at the control *U1* locus. qPCR reactions were performed in 10 µl in 384 well plates on a CFX-384 thermocycler (BioRad) with 5 µl of 2X PerfeCTa SYBR Green Supermix (Quantabio), 3 µl of DNA, and 200 nM final (2 µl) primer pair solutions. Each sample was done in technical triplicates. The following cycling conditions were used: 3 min at 95°C; 50 cycles: 15 s at 95°C, 30 s at 60°C, 30 s at 72°C. The amount of immunoprecipitated DNA at the *TLC1* locus was normalized to the *U1* locus and presented as the % of input.

## Acknowledgements

We thank the members of the Wellinger lab for constructive discussions and suggestions. We are very grateful to Emeline Pasquier, Gabriel St-Laurent, Erin Bonnell, Hannah Neumann, and Clara Diaz for their help with the revision experiments. We thank Pascal Chartrand, Erin Bonnell, and Emeline Pasquier for critical reading of the manuscript. We also thank Pascal Chartrand and Marlene Oeffinger labs for generously sharing *xpo1-1* and *mex67-5* strains. This work was supported by a grant from the Canadian Institutes of Health Research (RJW)(FDN154315); funding from the Centre de Recherche sur le Vieillissement (RJW); and a post-doctoral fellowship from the Fonds de Recherche du Quebec Santé to YV.

## Additional information

### Competing interests

Raymund J Wellinger: Reviewing editor, *eLife*. The other authors declare that no competing interests exist.

### Funding

| Funder | Grant reference number | Author |
|---|---|---|
| Canada Research Chairs | CRC in telomere biology | Raymund J Wellinger |
| Canadian Institutes of Health Research | FDN154315 | Raymund J Wellinger |
| Fonds de Recherche du Québec - Santé | Post-Doc Fellowship | Yulia Vasianovich |

The funders had no role in study design, data collection and interpretation, or the decision to submit the work for publication.

### Author contributions

Yulia Vasianovich, Conceptualization, Data curation, Formal analysis, Investigation, Visualization, Methodology, Writing - original draft; Emmanuel Bajon, Investigation, Methodology, Writing - review and editing; Raymund J Wellinger, Conceptualization, Resources, Data curation, Supervision, Funding acquisition, Project administration, Writing - review and editing

### Author ORCIDs

Yulia Vasianovich (iD) https://orcid.org/0000-0002-1643-794X
Emmanuel Bajon (iD) https://orcid.org/0000-0002-1588-2953
Raymund J Wellinger (iD) https://orcid.org/0000-0001-6670-2759

### Decision letter and Author response

Decision letter https://doi.org/10.7554/eLife.60000.sa1
Author response https://doi.org/10.7554/eLife.60000.sa2

## Additional files

### Supplementary files

• Supplementary file 1. Table of yeast strains used in the study. The table includes the names, genotypes and sources of all yeast strains used in the study. The yeast strains are referred to in the text using the names provided in the table.

• Supplementary file 2. Table of plasmids used in the study. The table includes the names, description and sources of all plasmids used in the study. The plasmids are referred to in the text using the names provided in the table.

• Supplementary file 3. Table of oligonucleotides used in the study. The table includes the names, sequences and description of all oligonucleotides used in the study. The oligonucleotides are grouped in the table based on their application. The oligonucleotides are referred to in the text using the names provided in the table.

• Transparent reporting form

### Data availability

Source data files have been uploaded. Strains and materials generated for this study will be freely available on request.

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
