## [Decision Letter]

**Acceptance summary:**

Using an induced Cre-dependent recombination system to replace expression of the endogenous *TLC1* gene by an MS2-tagged *TLC1* RNA, the authors show that the mRNA export receptor *Mex67* is essential for *TLC1* RNA biogenesis and stability of newly synthesized transcripts in the nucleus before they are exported by Xpo1, and that MS2-tagged the *TLC1* 3' mutant unable to bind the Sm_7_ complex crucial for processing are unstable and mainly detected in the cytoplasm, indicating that the Sm_7_-dependent processing steps occur in the cytoplasm. The manuscript provides new and relevant information to understand telomerase biogenesis related to a new role for the *Mex67* RNA export factor.

**Decision letter after peer review:**

Thank you for submitting your article "Telomerase biogenesis requires a novel *Mex67* function and a cytoplasmic association with the Sm_7_ complex" for consideration by *eLife*. Your article has been reviewed by three peer reviewers, and the evaluation has been overseen by Andrés Aguilera as the Reviewing Editor and James Manley as the Senior Editor. The following individuals involved in review of your submission have agreed to reveal their identity: Francoise Stutz (Reviewer #1); Vincent Geli (Reviewer #2).

The reviewers have discussed the reviews with one another and the Reviewing Editor has drafted this decision to help you prepare a revised submission.

Summary:

Yeast *TLC1* RNA biogenesis and processing have mostly been analyzed so far by examining the effect of a variety of mutations on endogenous steady state *TLC1* RNA pools. This new study uses an induced Cre-dependent recombination system to replace expression of the endogenous *TLC1* gene by an MS2-tagged *TLC1* RNA, so that they can distinguish the old untagged *TLC1* from the newly synthesized *TLC1*. This elegant approach allows a detailed description of the steps leading to the formation of a mature *TLC1* RNA in the nucleus, and finally to its degradation in the cytoplasm. The study confirms that the export receptor Xpo1 is the only one involved in the nuclear export of *TLC1* transcripts, which are then reimported via Kap122 but probably also other import receptors. Importantly, the authors show that the mRNA export receptor *Mex67* is essential for *TLC1* RNA biogenesis and stability of newly synthesized transcripts in the nucleus before they are exported by Xpo1. Finally, the authors show that MS2-tagged *TLC1* 3' mutant unable to bind the Sm_7_ complex crucial for processing are unstable and mainly detected in the cytoplasm, indicating that the Sm_7_-dependent processing steps occur in the cytoplasm and are essential for reimport into the nucleus.

Although several molecular details underlying *TLC1* biogenesis have still to be worked out, this study provides a novel more complete view of the *TLC1* RNA life cycle with well-designed and rigorously controlled experiments. The manuscript is very clearly written and referenced and is appropriate for publication, provided they respond to the questions raised below.

Essential revisions:

1) Several lines of evidence indeed suggest that *TLC1-MS2* is not fully functional: *TLC1-MS2* RNA is less abundant (Figure 1E), form less foci (Figure 2B) and is trapped in the cytoplasm for a long period after recombination (Figure 2E). Accordingly the telomeres are short in *TLC1-MS2* (Figure 2—figure supplement 1) and in contrast to cells expressing *TLC1-Sm2T* (Seto et al., 1999), *TLC1-Sm2T-MS2* cells senesce and eventually form type II survivors. Could the authors argue about this point?

2) The claim that the *Mex67* effect is independent of transcription regulation seems to rely basically on the use of thiolutin as a transcription blocker in an "epistasis" type of experiment. Since it is assumed that *TLC1* will get degraded in the nucleus in the *mex67-5* mutant, would it be possible to inactivate either the nuclear exosome (*Rrp6*) or *Rat1* and somehow more convincingly demonstrate that it is truly a degradation machinery that is responsible for the loss of *TLC1*. Alternatively the authors could try to demonstrate (through RNAPII ChIP) that transcription is not altered in the *mex67-5* mutant.

3) Could the authors demonstrate that the increased "new" nuclear *TLC1* in *xpo1-1* mutants is decreased in a *mex67-5* mutant.

---

## [Author Response]

Essential revisions:1) Several lines of evidence indeed suggest that TLC1-MS2 is not fully functional: TLC1-MS2 RNA is less abundant (Figure 1E), form less foci (Figure 2B) and is trapped in the cytoplasm for a long period after recombination (Figure 2E). Accordingly the telomeres are short in TLC1-MS2 (Figure 2—figure supplement 1).

This first part of point 1) questions whether tagging the *TLC1* RNA leads to a loss of function. As we mentioned and now elaborate in more detail (subsection “Tracking subcellular distribution of telomerase RNA fractions”, third paragraph), the tagged RNA does accumulate to a slightly lower level than the WT. We don’t know the reason for this but wish to emphasize that the slightly lower level of the RNA does not mean loss of functionality of the telomerase RNP. We carried out extensive control experiments on this issue (all published in Gallardo et al., 2011, and Bajon et al., 2015). In all assays, we were unable to detect any loss in functionality of the RNP per se, including telomerase enzymatic activity (Gallardo et al., 2011). Therefore, we concluded in these papers before and here too that the RNP per se is active as WT, but the RNA is somewhat less abundant, explaining the slightly shorter telomeres. For this manuscript here particularly important, the nucleo-cytoplasmic partitioning of the tagged RNA is exactly the same as for WT untagged RNA (see Figure 2—figure supplement 1E).

And in contrast to cells expressing TLC1-Sm2T (Seto et al., 1999), TLC1-Sm2T-MS2 cells senesce and eventually form type II survivors. Could the authors argue about this point?

Now, this second part of point 1) is a different issue. The reason why the cells in the referenced Seto/Cech study of ‘99 look like they maintain telomeres with the *tlc1-Sm2T* RNA molecule is twofold:

– First, Seto/Cech use a *tlc1Δ::LEU2* strain in which they express the various *TLC1* alleles, including the *tlc1-Sm2T* allele, from a plasmid borne gene copy. It is well known that plasmid-based gene expression results in an overexpression of the gene in question. Thus, they slightly overexpress the *tlc1-Sm2T* allele. All our genes are expressed from the genomic locus of *TLC1*.

– Secondly, the strain used in the Cech study is also *rad52Δ,* which thus cannot bypass telomerase problems with HR mediated telomere maintenance. We observe that *tlc1-Sm2T* expressing strains that are *RAD52wt* will senesce and form survivors (see Figure 5—figure supplement 1D). Hence, the apparent difference in the results is explained by higher expression and a strong selection of viable cells in the Seto et al., 1999, experiments. We actually think, but have not tested, that expressing the *tlc1-Sm2T* allele from the genomic locus in a *RAD52wt* strain would also yield survivors.

2) The claim that the Mex67 effect is independent of transcription regulation seems to rely basically on the use of thiolutin as a transcription blocker in an "epistasis" type of experiment. Since it is assumed that TLC1 will get degraded in the nucleus in the mex67-5 mutant, would it be possible to inactivate either the nuclear exosome (Rrp6) or Rat1 and somehow more convincingly demonstrate that it is truly a degradation machinery that is responsible for the loss of TLC1. Alternatively the authors could try to demonstrate (through RNAPII ChIP) that transcription is not altered in the mex67-5 mutant.

This indeed is an important point with great suggestions for experiments, thanks a bundle for that. We actually carried out both experiments. First, the experiment with a *mex67-5 rrp6Δ* double mutant strain at restrictive temperature bears out precisely what the reviewer predicted, namely complete suppression of the new RNA loss phenotype in the nucleus (new Figure 4C, E and Figure 4—figure supplement 1F-H). I am still amazed of how well this worked. Secondly, we did the RNA Pol II ChIP and compared Pol II loading on *TLC1* at permissive vs. restrictive temperature for the *mex67-5* cells. The results show pretty much equal Pol II loading and hence again fully support the notion that transcription is not affected in this condition (new Figure 4—figure supplement 1E). Both experiments are described in the subsection “The nuclear stability of new telomerase RNA transcripts requires *Mex67*”. Adding these two results to our story allows us to quite confidently conclude that the disappearance of the newly transcribed *TLC1* RNA molecules in the *mex67-5* cells at high temperature is due to degradation and not a loss of transcription.

3) Could the authors demonstrate that the increased "new" nuclear TLC1 in xpo1-1 mutants is decreased in a mex67-5 mutant.

In essence, the question is whether the *mex67-5ts* is epistatic to the *xpo1-1ts.* After much effort and way too much time, we actually were able to generate the double mutant. In brief, the result is exactly as the reviewer suspected: the *mex67-5ts xpo1-1ts* strain shows the same phenotype as the *mex67-5ts* alone and *mex67-5ts* thus is epistatic to *xpo1-1ts* (see new Figure 4—figure supplement 2A-C; subsection “The nuclear stability of new telomerase RNA transcripts requires *Mex67*”).